# Capture Concept through Comparison: Vision-and-Language Representation Learning with Intrinsic Information Mining

## Abstract

Achieving alignment between vision and language semantics poses a critical challenge. Prior works have sought to enhance alignment by incorporating additional supervision, such as tags or object bounding boxes, as anchors between modalities. However, these methods predominantly concentrate on aligning tangible entities, disregarding other crucial abstract *concepts* that elude perception, such as *side by side*. To overcome this limitation, we propose a novel approach to **C**apture various **C**oncepts through data **C**omparison (C3) for learning cross-modal representations. Specifically, we devise a data mining procedure to uncover intrinsic information within the database, avoiding the need for external annotations. Furthermore, we distinctly frame model inputs as triplets to better elucidate abstract semantics in images. Building upon this formulation, we propose two concept-centric pre-training objectives to signify concept learning. Extensive experiments demonstrate that models trained within the C3 framework consistently achieve significant enhancements across a wide range of comprehension and reasoning benchmarks, whether starting from scratch or fine-tuning from an existing model.

## 1 Introduction

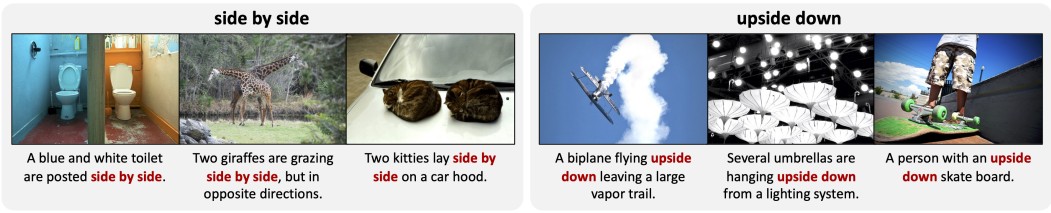

Figure 1: Examples of concepts mined from the database (Chen et al., 2015). The concepts could be abstract and shared across different scenes and subjects. The semantics of the concepts become clear when comparing multiple images carrying the same concept.

Semantic alignment between the domains of vision and language emerges as a crucial concern for various vision-language (VL) tasks. Consequently, numerous pre-training objectives have been meticulously designed to investigate the pairing relations between images and texts using large-scale datasets (Yao et al., 2022; Li et al., 2021a; Xue et al., 2021). However, the information in the two modalities is often inequivalent for most existing datasets. In other words, textual descriptions often fall short of providing a comprehensive account of each image (Yang et al., 2022). Such a weakly-aligned relation hinders the effective learning of cross-modal representations. Moreover, fine-grained alignments across modalities cannot be naturally achieved due to the lack of explicit annotations between entities and regions.

To alleviate this problem, prior works have sought to leverage additional supervision to bridge the gap between images and texts. For example, pre-trained object detectors are widely adopted. The detectors can be used to extract region-based features as visual inputs (Zhou et al., 2022; Li* et al., 2022; Kamath et al., 2021; Xu et al., 2021; Su et al., 2020), provide detected tags as additional

inputs to enhance alignments (Li et al., 2020; Zhang et al., 2021), or create learning targets for knowledge distillations (Liu et al., 2021). In addition to object detectors, other works attempt to obtain visual attributes in linguistic form through entity prompter (Li et al., 2022) or noun phrases of captions (Fang et al., 2022). However, prior efforts leveraging additional supervision still have limitations. Notably, these approaches focus on aligning data with concrete entities such as objects, regions, or attributes, which lack clear indications for aligning complex concepts that are challenging to precisely depict in the visual domain, such as "side by side" or "upside down" as shown in Fig. 1.

Another line of research, instead of relying on additional supervision, focuses on enhancing alignments through modifications to pre-training objectives or architectures (Fang et al., 2022; Kim et al., 2021; Xu et al., 2021; Yan et al., 2021; Huang et al., 2020). For instance, ALBEF (Li et al., 2021a) adopts an intermediate image-text contrastive loss to align the image and text features before performing cross-modal interactions in later layers. In addition to cross-modal alignments, TCL (Yang et al., 2022) further applies contrastive learning for intra-modal alignment by image or text augmentation. Nevertheless, most VL pretrained models still suffer from two issues. First, the supervision for alignments is limited in terms of diversity and quantity, often relying on the use of external models or predefined categories. Second, there is a lack of clear indications for learning concepts with abstract semantics, a critical requirement for tasks that demand comprehension and reasoning.

To tackle these challenges, we present a novel approach called **Capture Concept through Comparison (C3)**. The term "concept", rooted in psychology, is defined as *"the label of a set of things that have something in common"* (Archer, 1966). Inspired by the definition, we posit that a concept shall become more evident as more examples are provided. Therefore, the core idea of C3 is to leverage the data comparison to achieve concept-level alignments, thereby enhancing the comprehension of abstract semantics. To this end, we first propose a mining procedure to discover the concepts that are intrinsically shared among the database. Specifically, given an image-text pair, we extract text fragments and compare them with other texts in the training data. A fragment is identified as a concept if the same fragment appears in other texts. As such, this mining approach enables us to discover a broader spectrum of concepts without being constrained by external detectors or linguistic grammar.

Equipped with the mined concepts, the next challenge is to harness them for enhancing cross-modal alignment. An intuitive approach is to employ the concepts directly as language input and adhere to the conventional VL pre-training pipeline. However, we have discovered that employing an image-image-text triplet, i.e., two images with a concept text, can further assist models in discerning the abstract concept intertwined within images. Such a triplet formulation enables our model to learn a concept by pinpointing the "intersection" between two images, thereby streamlining the information to be focused in the visual domain and refining the alignment of concepts. With this input formulation, we design two concept-centric learning objectives, *Matched Concept Prediction (MCP)* and *Matched Concept Contrastive (MCC)*, to enhance alignments for both cross- and uni-modal representations. These objectives offer a direct learning mechanism for the mined concepts.

Finally, we assess our method under two configurations: continual pre-training and pre-training from scratch. The experimental results demonstrate that our approach can effectively leverage existing models without full re-training and significantly improve general VL behavioral testing. Furthermore, the experiments of pre-training from scratch highlight the benefits of concept-centric learning on various downstream tasks. Our main contributions can be summarized as follows: (i) We propose a novel mining procedure to discover the concepts intrinsic to the database, which is general and could potentially be leveraged in other studies as the immediate supervision for fine-grained alignments; (ii) We reformulate image-text learning scheme by considering image-image-text triplets, which facilitates models to identify and learn the abstract semantics in both modalities; (iii) We design two novel concept-centric objectives, i.e., Matched Concept Prediction (MCP) and Matched Concept Contrastive (MCC), to learn the matching of triples for better concept-level alignment; (iv) Extensive experiments and analysis demonstrate that the proposed concept-centric learning can improve both model capacity and downstream task performances.

## 2 RELATED WORK

Aligning vision and language representations is a critical challenge in VL pre-training. Recent works have attempted to address this challenge by leveraging additional supervision beyond traditional image-text pairs. For instance, Su et al. (2020) use Faster R-CNN to extract region of interest (RoI)

features, while Li et al. (2020) introduce object tags as the anchors for alignment. Zhang et al. (2021) improve on this approach by enhancing the visual representations via better pre-training of object detectors. Similarly, Xu et al. (2021) incorporate object detection objectives (Carion et al., 2020) into a sequence-to-sequence VL model, and Liu et al. (2021) rely on external detectors for object knowledge distillation. Other approaches include leveraging object detectors and phrase generators to learn hierarchical alignment (Li et al., 2022a) and performing contrastive learning with patch features and bounding boxes (Zeng et al., 2022). Additionally, Gao et al. (2022) propose aligning vision and language at different semantic levels, i.e., global image, local region, and ROI features for vision; summarization, caption, and attributes for language. Doveh et al. (2023) propose to teach VL models structure concepts by manipulating the text input based on pre-defined rules.

Another research line focuses on aligning solely with image-text pairs through modifications in objectives or architectures. Li et al. (2021a) introduce an intermediate image-text contrastive loss on unimodal features to facilitate subsequent cross-modal alignments. Other approaches suggest additional pre-training objectives, including word-region contrastive loss (Yao et al., 2023; Jiang et al., 2023), pseudo-labeled keyword prediction (Khan et al., 2022), weakly-supervised phrase grounding (Li et al., 2022b), token-wise maximum similarity (Yao et al., 2022), and visual dictionary as pixel-level supervision (Huang et al., 2021). Duan et al. (2022) encode features into a shared coding space defined by a dictionary of cluster centers for alignment. Yang et al. (2022) introduce intra-modal contrastive objectives to complement the cross-modal objectives. The integration of learning across vision, language, and multimodal tasks has been studied in Singh et al. (2022). A two-stage pre-training strategy has also been suggested in Dou et al. (2022a), involving initial coarse-grained training based on image-text data, followed by fine-grained training on image-text-box data.

In summary, the aforementioned approaches, aimed at enhancing alignments between vision and language, have showcased significant successes across a range of downstream tasks. Nevertheless, challenges persist in learning intricate concepts that resist easy specification through perceptual features. Although prior research has utilized additional supervision with respect to tangible entities like objects, regions, or attributes, these strategies are constrained when dealing with more abstract concepts. To overcome these limitations, we propose a framework that mines hidden concepts in the dataset and reformulates input based on the philosophy of the mining procedures, thereby enabling more effective alignment between vision and language.

## 3 METHOD

In this section, we propose a learning procedure for enhancing the fundamental abilities of the VL models, which can be effectively applied to both continual pre-training and pre-training from scratch. First, Sec. 3.1 describes the model architectures for better illustration. Sec. 3.2 and Sec. 3.3 introduce concept mining strategy and concept-centric objectives. Fig. 2 depicts an overview of C3.

### 3.1 OVERALL FRAMEWORK

C3 comprises a text encoder $\mathcal{E}_t$, an image encoder $\mathcal{E}_v$, and a cross-modal encoder $\mathcal{E}_{cross}$ as contemporary vision-language models (Kim et al., 2021; Li et al., 2021a; Zhang et al., 2021). We adopt such a succinct architecture and focus on studying the proposed concept-centric pre-training. A text $T$ is tokenized into a sequence of subwords $[t_1, t_2, ...]$, and two special tokens $t_{cls}$ and $t_{sep}$ are respectively prepended and appended to the sequence. The sequence is then passed through $\mathcal{E}_t$ to obtain the unimodal features. An image $I$ is first divided into several patches and processed by a convolutional layer to extract patch features $[v_1, v_2, ...]$. These patch features are then flattened and fed into the $\mathcal{E}_v$ for further feature extraction. We also add a learnable vector $v_{cls}$ to aggregate global information for the vision modality. For fusing the features from unimodal encoders, we apply co-attention modules (Dou et al., 2022b) as cross-modal encoders $\mathcal{E}_{cross}$ for both vision and language. Finally, for an image-text pair, the vision-language joint representation $z$ is obtained as follows:

$$H_t = [h_{cls}^t, h_1^t, ...] = \mathcal{E}_t([t_{cls}, t_1, ...]), H_v = [h_{cls}^v, h_1^v, ...] = \mathcal{E}_v([v_{cls}, v_1, ...]),$$
$$Z = [z_{cls}^t, z_1^t, ..., z_{cls}^v, z_1^v, ...] = \mathcal{E}_{cross}([H_t, H_v]), z = [z_{cls}^t, z_{cls}^v]. \tag{1}$$

We pre-train C3 from scratch with the proposed concept-centric objectives (Section 3.3), i.e., Matched Concept Prediction (MCP) or Matched Concept Contrastive (MCC), and the widely-used pair-centric objectives, such as Image Text Matching (ITM) and Masked Language Modeling

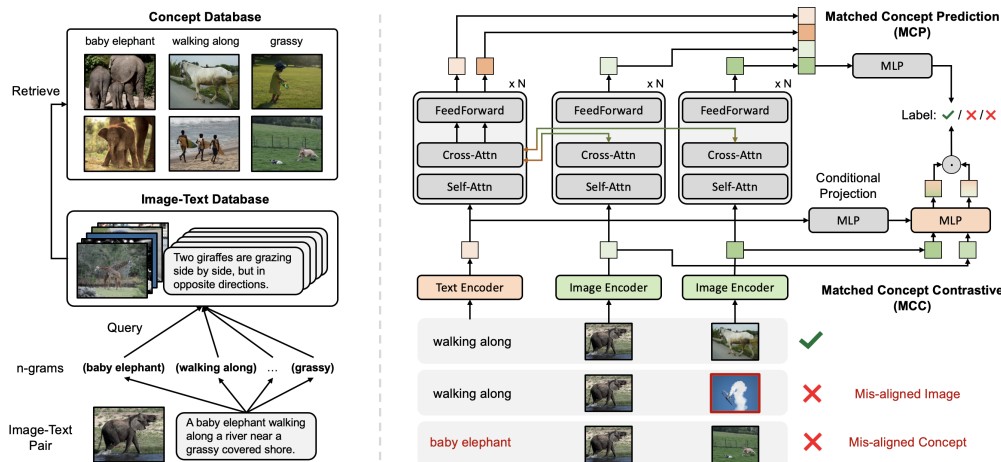

Figure 2: The proposed concept mining procedure (left) and concept-centric pre-training architecture and objectives (right). As shown on the left side, we use the n-grams from the text as the queries to retrieve images also containing the n-grams in their texts. With the mined n-gram concepts, the inputs are formulated as triplets for pre-training, shown on the right side.

(MLM). For the continual pre-training, we insert trainable low-rank residual adapters (LoRA (Hu et al., 2022)) into existing models and learn with the MCC objective to further enhance the capacity.

## 3.2 CONCEPT-CENTRIC LEARNING FORMULATION

**Mutual Information Maximization**. Existing works (Chen et al., 2020; Dou et al., 2022b; Kim et al., 2021; Li et al., 2021a; 2020; Su et al., 2020; Zhang et al., 2021) commonly adopt the combinations of Masked Language Modeling (MLM), Masked Vision Modeling (MVM), and Image-Text Matching/Contrastive (ITM/ITC) for VL pre-training. Previous work (Li et al., 2021a) shows that these objectives can be interpreted as the maximization of the mutual information (MI) between different views of an image-text pair. For example, ITC treats the image and text as two different views; MLM treats the masked tokens as one view and other tokens with the image as another view. In other words, these approaches aim to learn the multimodal representations invariant across different views for improving downstream tasks. It is noteworthy that the aforementioned approaches maximize the MI for each image-text pair independently. Besides, the views are considered either at the instance-level (ITC) or token-/patch-level (MLM/MVM). However, we argue that models can be better learned by considering *concepts* that are cross-data and diverse in granularity. Therefore, instead of considering only a single image-text pair, we construct a novel learning formulation by centering a concept that is built from multiple image-text pairs.

Specifically, we define two random variables $c_1$ and $c_2$ as two different views of a concept, where the views correlate to *a concept text* and *multiple images*. We could maximize a lower bound on $\text{MI}(c_1, c_2)$ by minimizing the InfoNCE loss (Li et al., 2021a; Oord et al., 2018) defined as follows.

$$\mathcal{L}_{\text{NCE}} = -\mathbb{E}_{p(c_1, c_2)} \log \frac{\exp(f(c_1, c_2))}{\sum_{c_2' \in B} \exp(f(c_1, c_2'))}, \quad (2)$$

where $f(\cdot)$ is a scoring function and $B$ is a batch containing one positive sample with other negative samples. To realize this learning framework, we next elaborate on the proposed methods for mining concepts in the database.

---

**Algorithm 1:** Concept Mining

**Data:** Image-text database $\mathcal{D} = \{(I_i, T_i)\}_{i=1}^{N_\mathcal{D}}$.
**Result:** Concept database $\mathcal{D}^c = \{(I_i, \mathcal{I}_i, \mathcal{C}_i)\}_{i=1}^{N_\mathcal{D}}$.
**for** $(I_i, T_i) \in \mathcal{D}$ **do**
    Initial $\mathcal{C}_i$ and $\mathcal{I}_i$ as empty sets; $k = 0$;
    Obtain $n$-grams $\mathcal{G}$ from $T_i$ for $n \in [N, ..., 1]$;
    **for** $G \in \mathcal{G}$ **do**
        Random sample a subset $\mathcal{D}_s$ from $\mathcal{D}$;
        Initial $\mathcal{I}_k$ as empty sets;
        **for** $(I_j, T_j) \in \mathcal{D}_s$ **do**
            **if** $G \in T_j$ and $|\mathcal{I}_k| \le K_1$ **then**
                Assign $G$ to $C_k$; Add $I_j$ to $\mathcal{I}_k$;
        **end**
        **if** $\mathcal{I}_k$ *is not empty and* $|\mathcal{I}_i| \le K_2$ **then**
            Add $C_k$ to $\mathcal{C}_i$; Add $\mathcal{I}_k$ to $\mathcal{I}_i$; $k$ += 1;
    **end**
**end**

---

**Concept Mining**. Drawing inspiration from the field of psychology, which defines a concept "as the label of a set of things that have something in common" (Archer, 1966), we propose to mine the concepts by exploring the commonality between pairs of data. Our approach is based on the identification of *overlapping n-grams* (Järvelin et al., 2007; Wang et al., 2007) between pairs of data, specifically image-text pairs in a database $\mathcal{D} = \{(I_i, T_i = \{t_{i1}, t_{i2}, ...\})\}_{i=1}^{N_\mathcal{D}}$. In each iteration $i$, we extract all $n$-grams in the associated text $T_i$ as $\{(t_{i1}, ..., t_{in}), (t_{i2}, ..., t_{in+1}), ...\}$. Next, we treat each $n$-gram as a query to retrieve images carrying the same $n$-gram in their texts. If there is any matching, the $n$-gram is defined as a concept shared across these data. We retrieve at most $K_1$ pairs for a concept and early terminate the current iteration if $K_2$ pairs are obtained for $T_i$. To allow for concepts of varying granularity, we consider different $n \in [1, N]$ and mine the concepts with a descending order of $n$ since the longer concept covers the shorter one. In the end, each image may involve multiple concepts, and each concept correlates to multiple different images. Let $\mathcal{C}_i$ and $\mathcal{I}_{ik} = \{I_{ikj}\}$ respectively denote the matched concept set for $i$-th sample and the matched image set of $k$-th concept $\mathcal{C}_{ik}$. Accordingly, the concept database is constructed as follows:

$$\mathcal{D}^c = \{\{(I_i, \mathcal{I}_{ik}, \mathcal{C}_{ik})\}_{k=1}^{|\mathcal{C}_i|}\}_{i=1}^{N_\mathcal{D}}. \tag{3}$$

The mining procedure is further presented in Algorithm 1.

### 3.3 Concept-centric Pre-training

Based on the concept database, we propose to maximize the mutual information across an n-gram concept and two corresponding images. The input formulation is therefore transformed from pair-based (i.e., image-text) to triplet-based (i.e., image-image-text). In the following, we present two concept-centric objectives to explore this formulation.

**Matched Concept Prediction (MCP).** Different from prior works, MCP takes a concept text and a pair of images as the input. This objective aims to predict whether the concept $C$ is shared between the two images $(I_i, I_j)$, which provides explicit supervision to learn the semantics of concepts across modalities. An image could encapsulate numerous concepts in different granularity, and the contrasts of two images help capture and identify the specified concept more efficiently. For a triplet $(I_i, I_j, C)$, we divide it into $(I_i, C)$ and $(I_j, C)$ to encode them respectively. Let $z_{ij}^c$ denote the concatenation of joint representations from $(I_i, C)$ and $(I_j, C)$. To obtain the negative examples for learning, we investigate two strategies. The first one is to replace one of the images in a positive triplet with a mismatched image $I_j'$, i.e., $(I_i, I_j', C)$, while the other is to replace the concept with another concept $C'$, i.e., $(I_i, I_j, C')$. As such, we could define the objective as:

$$\widehat{\mathcal{L}}_{\text{MCP}} = -\mathbb{E}_{p(I_i, I_j, C)}[\log \frac{\exp(\psi^\top z_{ij}^c)}{\sum_{(I_j', C') \in B} \exp(\psi^\top z_{ij'}^{c'})}], \tag{4}$$

where $\psi$ is a learnable matrix. However, since the objective utilizes multimodal representations, it requires forwarding all triplets in a batch independently despite some images or concepts being shared, making the optimization memory-intensive in practice. Therefore, we adopt the local NCE (Gutmann & Hyvärinen, 2010; Gutmann & Hyvärinen, 2012) to approximate the loss (Kong et al., 2020; Liu et al., 2022) as:

$$\mathcal{L}_{\text{MCP}} = -\mathbb{E}_{p(I_i, I_j, C)}[y_{ij}^c \log \phi_{\text{MCP}}(z_{ij}^c) + (1 - y_{ij}^c) \log(1 - \phi_{\text{MCP}}(z_{ij}^c))], \tag{5}$$

where $y_{ij}^c$ is the label and $\phi_{\text{MCP}}$ is a network producing a value as probability. This formulation in another way leverages a binary classifier to distinguish matched samples from the noisy ones.

**Matched Concept Contrastive (MCC)**. Apart from utilizing cross-modal representations for learning the alignment of concepts, we could extend such an idea with the unimodal ones. Specifically, we use the outputs of the image and text encoders to learn the matching of triplets before the cross-modal layers. This strategy could be beneficial for the cooperated objectives to leverage the aligned representation in an early stage. The objective is presented as:

$$\mathcal{L}_{\text{MCC}} = -\mathbb{E}_{p(I_i, I_j, C)}[\log \frac{\exp(s(\psi_c^\top h_i, \psi_c^\top h_j))}{\sum_{(I_j', C') \in B} \exp(s(\psi_{c'}^\top h_i, \psi_{c'}^\top h_{j'}))}], \tag{6}$$

where $s(\cdot)$ is cosine similarity, $h_i = h_{cls}^{I_i}$, $\psi_c = \phi_{\text{MCC}}(h_{cls}^C)$, and $\phi_{\text{MCC}}$ is an MLP-based network. In this formulation, we use the concept to generate a projection matrix $\psi_c$ that transforms the two images into a space that is conditioned on the concept. This operation is inspired by the general conditioning methods proposed in Perez et al. (2018).

## 4 EXPERIMENTS

### 4.1 EXPERIMENTAL SETTINGS

**Datasets and Benchmarks**. We conduct pre-training on four image-caption datasets: COCO (Chen et al., 2015), Visual Genome (VG) (Krishna et al., 2017), Conceptual Captions (CC) (Sharma et al., 2018), and SBU Captions (Ordonez et al., 2011). Our model evaluations encompass a range of vision-language benchmarks, including vision-language behavior assessment (VL-Checklist (Zhao et al., 2022)), visual entailment (SNLI-VE (Xie et al., 2019)), natural language visual reasoning (NLVR$^2$(Suhr et al., 2019)), and image-text retrieval (Flickr30k(Plummer et al., 2015)).

**Training Configurations.** We evaluate our methods in two configurations: *continual pre-training* and *pre-training from scratch*. In the case of continual pre-training, our goal is to assess the advantages of applying concept-centric learning to existing models without necessitating a full model re-training. Leveraging pre-trained knowledge can prove both effective and cost-efficient. Specifically, we select CLIP (Radford et al., 2021) as our base model, which has undergone pre-training on 400 million image-text pairs and has been applied to a wide range of tasks. In this context, we introduce LoRA (Hu et al., 2022) to enhance the base model's capacity while keeping all base model parameters fixed, allowing only the parameters of LoRA to be trainable. For the pre-training from scratch scenario, we follow the setup of METER (Dou et al., 2022b), given its relatively manageable pre-training scale, utilizing 4.0 million images and 5.1 million image-text pairs for pre-training.

**Implementation Details.** For continual pre-training, we initialize the model with CLIP-ViT-B/32 or CLIP-ViT-B/16 and train it for 1 epoch using COCO, VG, CC, and SBU, respectively. The trainable parameters constitute approximately 1.2% of the entire model. The concept mining procedure is executed within each dataset, considering $n$-grams with $n$ ranging from 1 to 5 as concept candidates. To ensure comprehensive coverage, we set the hyperparameters $K_1$ and $K_2$ to 5 and 80, respectively. In the case of pre-training from scratch, our model undergoes training for 50k steps on COCO, VG, CC, and SBU, which is half the number of learning steps compared to our baseline METER (Dou et al., 2022b) (pre-trained with 100k steps). For ablation and analysis purposes, we train the models from scratch for 2.6k steps on COCO, enabling extensive experimentation. All images are resized to 224×224 through center-cropping during the pre-training process. [1]

### 4.2 VISION-LANGUAGE BEHAVIORAL TESTING

We first assess the fundamental vision-language capability of C3 from different angles with the VL-Checklist benchmark. The comparison with the base model under different configurations is shown in Table 1. The results reveal that directly continuing the pre-training may deteriorate performance, while our methods can significantly and consistently improve the VL capability across diverse aspects. Notably, the enhancement in the object aspect is evident across various data sources. This could result from the fact that most concepts would naturally involve objects, contributing more to this aspect. Besides, the improvement in the attribute aspect is particularly pronounced in COCO, signifying that COCO includes more attributed-related descriptions, such as the size, color, and material, which C3 can effectively identify and leverage. VG includes abundant region descriptions, which enables C3 to grab the concept of objects and their relations by comparing data even without using the structured annotations in this dataset, which is hard to learn by image-caption pairs. The diverse characteristics of data sources also underscore the idea that different data sources may cover different concepts, which can be successfully exploited by C3 to enhance the model's overall capabilities. Importantly, our methods prove beneficial for both continual pre-training and pre-training from scratch settings, highlighting the generalizability of the proposed approach.

### 4.3 VISION-LANGUAGE REASONING

To compare with prior works, C3 uses the same framework architecture as our primary baseline model, METER. Given the resource-intensive nature of VL pre-training, learning efficiency is of utmost importance. Therefore, we compare C3 with models of similar data scales and learning steps. Our goal is to achieve state-of-the-art performance with fewer training steps, as the model tends to improve with increased data and learning steps (Rae et al., 2021). As shown in Table 2, C3 attains

---

[1]More implementation details for pre-training and fine-tuning are provided in the appendix.

Table 1: Performance comparison on VL-Checklist (Zhao et al., 2022). *C/V/O/S* refers to the CC/VG/COCO/SBU dataset. For CLIP architecture, the model w/o C3 is an ablation of MCC.

| Base Model | Dataset | Attribute w/o C3 → w/ C3 | Object w/o C3 → w/ C3 | VL-Checklist Relation w/o C3 → w/ C3 | Average w/o C3 → w/ C3 | Δ w/o C3 → w/ C3 |
|---|---|---|---|---|---|---|
| | | Continual Pre-training | | | | |
| CLIP-ViT-B/32 | - | 69.09 | 81.94 | 63.30 | 71.44 | - |
| | S | 69.09 → 70.87 | 81.24 → 81.99 | 60.62 → 63.68 | 70.32 → 72.18 | -1.12 → 0.74 |
| | C | 68.63 → 71.18 | 80.32 → 82.04 | 53.75 → 55.72 | 67.57 → 69.65 | -3.87 → -1.79 |
| | V | 71.59 → 72.31 | 87.28 → 87.16 | 62.86 → **64.77** | 73.91 → **74.75** | 2.47 → 3.31 |
| | O | 71.43 → **75.26** | 85.38 → **87.36** | 57.66 → 61.11 | 71.49 → 74.58 | 0.05 → 3.14 |
| CLIP-ViT-B/16 | - | 70.37 | 82.94 | 61.98 | 71.76 | - |
| | S | 70.68 → 71.13 | 82.85 → 83.40 | 61.52 → 62.64 | 71.68 → 72.39 | -0.08 → 0.63 |
| | C | 69.66 → 70.18 | 81.54 → 87.56 | 55.86 → 62.86 | 69.02 → 73.53 | -2.74 → 1.77 |
| | V | 68.40 → 69.98 | 87.34 → 87.75 | 60.80 → **62.98** | 72.18 → 73.57 | 0.42 → 1.81 |
| | O | 71.30 → **75.87** | 86.94 → **88.48** | 53.98 → 61.75 | 70.74 → **75.37** | -1.02 → 3.61 |
| | | Pre-training from Scratch | | | | |
| METER | C+V+O+S | 81.65 → **84.28** | 84.72 → **89.04** | 71.94 → **73.90** | 79.44 → **82.41** | 2.97 |

Table 2: Performance comparison on SNLI-VE (Xie et al., 2019) and NLVR[2] (Suhr et al., 2019).

| Model | Images | Iters | Params | SNLI-VE dev | test | NLVR[2] dev | test |
|---|---|---|---|---|---|---|---|
| ALBEF(14M) (Li et al., 2021a) | 14M | 420M | 500M | 80.80 | 80.91 | 82.55 | 83.14 |
| SimVLM_HUGE (Wang et al., 2022) | 1.8B | 4.1B | 632M | 86.21 | 86.32 | 84.53 | 85.15 |
| BEIT-3 (Wang et al., 2023) | 36M | 6.1B | 1.9B | - | - | 91.51 | 92.58 |
| PixelBERT (Huang et al., 2020) | 207K | 8.3M | 170M | - | - | 76.5 | 77.2 |
| Visual Parsing (Xue et al., 2021) | 221K | 8.8M | 308M | - | - | 77.61 | 78.05 |
| OSCAR_LARGE (Li et al., 2020) | 4M | 512M | 380M | - | - | 79.12 | 80.37 |
| ViLT (Kim et al., 2021) | 4M | 819M | 114M | - | - | 75.70 | 76.13 |
| UNITER_LARGE (Chen et al., 2020) | 4M | - | 343M | 79.39 | 79.38 | 79.12 | 79.98 |
| VILLA_LARGE (Gan et al., 2020) | 4M | - | 343M | 80.18 | 80.02 | 79.76 | 81.47 |
| UNIMO_BASE (Li et al., 2021b) | 5.7M | 1.5B | 165M | 80.00 | 79.10 | - | - |
| VinVL_BASE (Zhang et al., 2021) | 5.7M | 1B | 290M | - | - | 82.05 | 83.08 |
| CLIP-ViL (Shen et al., 2022) | 4M | 184M | 330M | 80.61 | 80.20 | - | - |
| ALBEF(4M) (Li et al., 2021a) | 4M | 154M | 500M | 80.14 | 80.30 | 80.24 | 80.50 |
| METER (Dou et al., 2022b) | 4M | 410M | 384M | 80.86 | 81.19 | 82.33 | 83.05 |
| C3 (our) | 4M | 205M | 384M | **81.30** | **81.34** | **82.36** | **83.35** |

superior performance in NLVR[2] and SNLI-VE, requiring fewer training iterations than METER, and surpasses ALBEF. These results suggest that learning concepts through image comparison can enhance a model's inference capabilities.[2]

## 4.4 ABLATION STUDY

**Pre-training Objectives.** We investigate diverse pre-training settings by restricting the learning steps to 2.6k and using CLIP-ViT-224/32 as the vision encoder. Table 3 demonstrates that incorporating MCP (row 4) enhances all tasks compared to the METER baseline (row 1), particularly for image-text retrieval and the VL-Checklist, owing to the improved representations learned with multi-grained concept alignment. Furthermore, our approach naturally meets the requirements of NLVR[2], where models must assess the accuracy of descriptions between two images, bringing in additional benefits. Notably, the ablation study demonstrates that ITM is critical for retrieval tasks and the VL-Checklist but not for semantic inference tasks (row 3), while MLM greatly impacts the performance of the reasoning task (row 2). Thus, each objective covers distinct aspects, and the best performance can be achieved by combining them.

**Training Sample Formation.** We investigate the impacts of two methods for constructing negative samples of MCP (row 4 & row 5-6). We refer to the misaligned image method as *type-1-negative* and the misaligned concept method as *type-2-negative*. Results indicate that type-2-negative is relatively

---

[2]Additionally, we evaluate C3 on the zero-shot and fine-tuned image-text retrieval tasks to assess cross-modal representation quality, as shown in the appendix.

Table 3: Ablation study of C3. The first row is METER (Dou et al., 2022b). All models are trained for 2.6k steps on 224×224 images with patch size 32. *I/C* refers to constructing negative samples by misaligned images/concepts. *PW* refers to the pairwise formulation for the MCP objective.

| ITM | MLM | MCP | VL-Checklist Att / Obj / Rel | Δ | Flickr30K-ZS IR / TR | Δ | NLVR² dev / test | Δ | SNLI-VE dev / test | Δ |
|---|---|---|---|---|---|---|---|---|---|---|
| | | | n-gram | | | | | | | |
| ✓ | ✓ | - | 54.13 / 73.45 / 54.80 | -6.2 | 55.82 / 62.17 | -5.2 | 72.83 / 74.35 | -1.2 | 76.98 / 77.31 | -0.5 |
| ✓ | - | ✓ | 62.57 / **77.70** / 53.20 | -2.5 | **60.55** / 67.49 | -0.1 | 69.47 / 71.47 | -4.4 | 76.41 / 76.58 | -1.1 |
| - | ✓ | ✓ | 50.70 / 46.77 / 45.38 | -19.4 | 2.95 / 6.23 | -59.6 | 73.79 / 74.78 | -0.6 | **77.62** / 77.57 | 0.0 |
| ✓ | ✓ | ✓ | **64.52** / 76.53 / **60.05** | 0.0 | 60.45 / **67.87** | 0.0 | **74.27** / **75.40** | 0.0 | 77.55 / **77.67** | 0.0 |
| ✓ | ✓ | I | 61.21 / 76.98 / 57.86 | -1.7 | 58.98 / 64.87 | -2.2 | 74.01 / 74.49 | -0.6 | 77.31 / 77.53 | -0.2 |
| ✓ | ✓ | C | 61.66 / 77.30 / 57.26 | -1.6 | 60.07 / 66.20 | -1.0 | 73.33 / 74.52 | -0.9 | 77.47 / 77.66 | 0.0 |
| ✓ | ✓ | PW | 61.44 / 77.48 / 58.23 | -1.3 | 58.87 / 65.67 | -1.9 | 73.91 / 74.90 | -0.4 | 77.16 / 77.46 | -0.1 |
| | | | noun phrase | | | | | | | |
| ✓ | ✓ | ✓ | 62.03 / 77.22 / 53.74 | -2.7 | 58.50 / 64.53 | -2.6 | 73.97 / 75.25 | -0.2 | 77.23 / 77.52 | -0.2 |

effective since it is more challenging, forcing models to learn semantics without relying on spurious clues. Comparatively, type-1-negative is formed by replacing the matched image with a random one, where the image pairs mostly do not have clearly shared concepts. Therefore, models might be able to make predictions solely by comparing visual features. Nevertheless, the combination of both types still yields the best performance. Furthermore, to understand the effectiveness of the triplet input formulation, we compare it with pairwise input. By altering the input from triplet to pairwise, the MCP objective aims to predict whether a concept is present in an image. The results (row 7) show that triplet training still outperforms pairwise training across all metrics. This is likely because an image can contain multiple concepts, making direct alignment via pairwise training ambiguous and inefficient. In contrast, triplet training explicitly provides two references for each mined concept, reducing the number of potential concepts to be considered in the visual domain and enabling more precise alignment. Additionally, pairwise training still improves upon the baseline model (row 1), highlighting the efficacy of learning with concepts.

**Concept Mining Strategy.** We propose extracting overlapped n-grams to form concepts, which can identify a wider range of concepts compared to previous works limited to specific scopes such as object tags (Li et al., 2020) or verb-/adj-nouns (Kamath et al., 2022). To evaluate the benefits, we learn a baseline model on a restricted concept database by performing the proposed mining procedure but only considering noun phrases. As shown in the bottom row of Table 3, the results demonstrate that our approach (row 4) outperforms the baseline across all tasks, indicating that the proposed mining procedure is critical and can serve as a general method for exploring concepts in learning.

## 4.5 ANALYSIS FOR DIFFERENT SEMANTIC COMPLEXITY

The proposed C3 model aligns concepts to enhance inter-modality semantic relationships, thereby improving its reasoning capabilities for complex semantics. To validate the claim, we analyze the models' performance under various levels of semantic complexity in the challenging NLVR² visual reasoning task. We define the semantic complexity of data from three perspectives, i.e., the constituency parsing tree, character length, and token length. The maximum depth of the constituency parsing tree is selected as the measure of semantic complexity since deeper trees generally indicate more intricate text structures, while longer text or word lengths suggest richer contexts. Fig. 3 consistently demonstrates that the C3 model's performance is negatively correlated with semantic complexity across all three definitions and experimental settings. Furthermore, our proposed

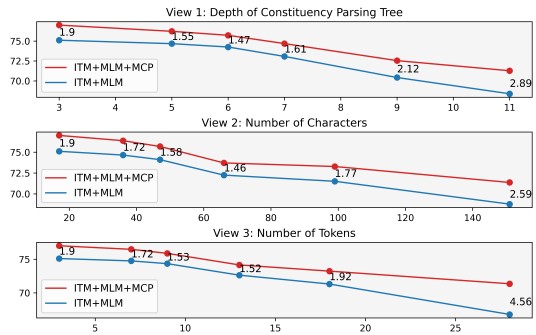

Figure 3: The performance of NLVR² under varying levels of semantic complexity.

matched concept prediction shows to be particularly effective for challenging instances, highlighting its practical value.

## 4.6 VISUALIZATION

For a more intuitive understanding, we visualize the attention maps generated by the final attention layer within the cross-modal encoder. To gain insights into the models' comprehension of challenging concepts, we randomly select concepts with 4- or 5-gram attributes from the validation dataset. Figure 4 displays these visualizations for both C3 and our baseline model, METER. Notably, the results illustrate that C3 exhibits the ability to focus on specific regions in accordance with the text fragments, while METER tends to distribute attention more evenly across the entire space. This divergence may arise from the difficulty in perceiving sentence fragments due to distribution differences between concept texts and captions. However, through concept-specific learning, models find it easier to identify meaningful regions for sentence fragments, thus enhancing their overall capacity.

Examples a & b, as well as examples c & d, demonstrate that C3 adapts its attention to different regions according to the input concepts. Furthermore, C3 outperforms METER in appropriately attending to regions for ambiguous phrases, as exemplified in examples e, f, and g. These visualizations suggest that C3 holds significant potential for tasks involving visual-linguistic grounding (Kazemzadeh et al., 2014; Mao et al., 2016; Yu et al., 2016), opening up promising avenues for future research. Additional examples can be found in the appendix.

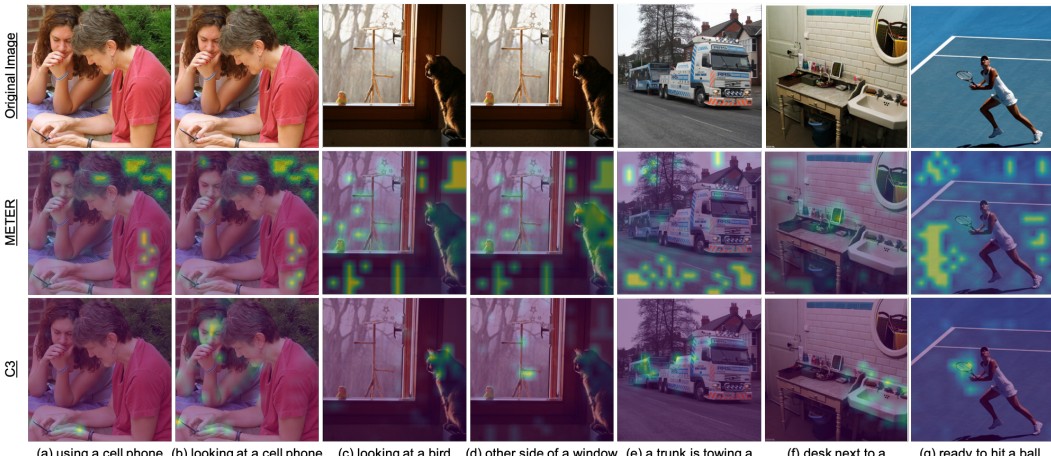

Figure 4: Visualization of attention maps for abstract concepts randomly selected from examples containing 4- or 5-gram concepts, providing insight into the model's understanding of abstract semantic relationships. Examples a & b and examples c & d show that C3 can attend to different regions depending on the input concepts. Examples e, f, and g show that C3 can process ambiguous sentence fragments.

## 5 CONCLUSION

This paper presents Capture Concepts through Comparison (C3), a novel framework designed to enhance the core capabilities of vision-language (VL) models by strengthening the semantic alignment between the realms of vision and language. To begin, we introduce a mining procedure aimed at uncovering latent concepts within the database, all without the need for predefined scopes or external annotations. Building upon these mined concepts, we put forth two innovative learning objectives tailored for different architectural choices for the model, where inputs are formulated as triplets comprising a concept and images. These settings align with the psychological insight that concepts are shaped by shared characteristics. Finally, our comprehensive experiments conclusively demonstrate that C3 effectively boosts model capacity and enhances performance on downstream tasks, both in the context of continual pre-training and pre-training from scratch. These findings underscore the efficacy and versatility of our concept-centric learning approach.

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

## A  IMPLEMENTATION DETAILS

**Pre-training Dataset.** Table 4 shows the statistics of our pre-training datasets. VG and CC have more concepts than the others due to their larger number of captions. The average length of concepts reflects the diversity across databases, where a shorter one could suggest less overlapping tokens or more diverse images.

Table 4: The statistics of concept database $\mathcal{D}^c$.

|  | COCO | VG | SBU | CC |
|---|---|---|---|---|
| # Images | 113K | 108K | 867K | 3.01M |
| # Captions | 567K | 5.41M | 867K | 3.01M |
| # Unique concepts | 494K | 742K | 513K | 965K |
| Avg. length of concepts | 3.98 | 4.62 | 3.71 | 3.89 |
| Avg. # of concepts per image | 19.71 | 22.02 | 3.43 | 3.14 |
| Avg. # matched image per image | 59.26 | 82.63 | 10.22 | 9.19 |

**Pre-training Settings for Improving Off-the-shelf Models.** We choose CLIP (Radford et al., 2021) as our base model initialized by the pre-trained weights from OpenAI[3]. To effectively leverage the pre-trained knowledge, we finetune the CLIP models by inserting parameter-efficient finetuning layers, LoRA (Hu et al., 2022; Mangrulkar et al., 2022), and optimize the models with the contrastive learning objective and MCC objective jointly. The dimension of the low-rank matrics is set to 8, and the scaling factor is 32. The dropout probability of the LoRA layers is 0.1.

**Pre-training Settings for Comparing with Prior Arts.** We utilize a 12-layer language transformer initialized with RoBERTa-base (Liu et al., 2019) as the text encoder, denoted by $\mathcal{E}_t$. Similarly, we employ a 12-layer visual transformer initialized with CLIP-ViT-224/16 (Dosovitskiy et al., 2021) as the image encoder, denoted by $\mathcal{E}_v$. To combine the features from the single-modal encoders, we utilize 6-layer co-attention modules (Dou et al., 2022b) as cross-modal encoders $\mathcal{E}_{cross}$ for both vision and language. The C3 model has 396M pre-training parameters in total. We jointly train the model with ITM, MLM, and MCP for 50k steps with batch size 4096. We adopt the AdamW (Loshchilov & Hutter, 2019) and pre-train on COCO, VG, SBU, and CC. During pre-training, all images are resized to $224 \times 224$ through center-cropping. The pre-training learning rates for the single-modal and cross-modal layers are respectively set to 1e-5 and 5e-5, and the warm-up ratio is set to 10%. The default masking ratio of MLM is 0.3.

**Pre-training Settings for Ablation Study and Analysis.** For ablation and analysis purposes, we pre-train the models for 2.6k steps on COCO to facilitate extensive experimentation. The default image encoder $\mathcal{E}_v$ is initialized with CLIP-ViT-224/32 (Dosovitskiy et al., 2021).

**Fine-tuning Settings.** The learning rate of Flickr30K fine-tuning is set to 5e-6 with image size 288 and trained for 5 epochs. The learning rate of NLVR fine-tuning is set to 1e-5 with image size 288 and trained for 10 epochs. The learning rate of SNLI-VE fine-tuning is set to 2e-6 with image size 384 and trained for 5 epochs. The learning rate of VQAv2 fine-tuning is set to 5e-6 with image size 512 and trained for 10 epochs. The warm-up ratios for all experiments are set to 10%. We notice that increasing the image size is more critical for the visual question-answering task than for the image-text retrieval and reason tasks.

---

[3]We initialize the CLIP-Vit-B/32 with https://huggingface.co/openai/clip-vit-base-patch32, and initialize CLIP-Vit-B/16 with https://huggingface.co/openai/clip-vit-base-patch16.

## B  PERFORMANCE ON DOWNSTREAM TASKS

Table 5 and Table 6 shows that C3 achieves exceptional performance with fewer training iterations on retrieval tasks, indicating that the concept-centric learning framework contributes to cross-modal feature fusion. Notably, despite ALBEF's designed objectives for retrieval tasks, C3 performs comparably. Table 7 presents the performance comparison of C3 and various baselines on VQAv2. The results reveal that C3 performs on par with METER. We attribute this similarity to the comparatively lower demands for abstract or complex concept understanding in VQAv2 questions. Besides, we do not perform a hyper-parameter search for C3 during VQAv2 finetuning, while METER reported results obtained through grid searches over the learning rates {1e-6,2e-6,5e-6,1e-5} and a higher image 576. Thus, To ensure a fair comparison and better understand the advantage of C3, we conduct an ablation study in the next section, where all models are trained with identical configurations.

Table 5: Performance comparison of fine-tuned image-text retrieval on Flickr30K.

| Model | Images | Iters | Params | Flickr30K | | | | | |
| | | | | Image Retrieval | | | Text Retrieval | | |
| | | | | R@1 | R@5 | R@10 | R@1 | R@5 | R@10 |
|---|---|---|---|---|---|---|---|---|---|
| ALBEF(14M) (Li et al., 2021a) | 14M | 420M | 500M | 85.6 | 97.5 | 98.9 | 95.9 | 99.8 | 100.0 |
| BEIT-3 (Wang et al., 2023) | 36M | 6.1B | 1.9B | 90.3 | 98.7 | 99.5 | 98.0 | 100.0 | 100.0 |
| SOHO (Huang et al., 2021) | 221K | 8.8M | 84M | 72.5 | 92.7 | 96.1 | 86.5 | 98.1 | 99.3 |
| Visual Parsing (Xue et al., 2021) | 221K | 8.8M | 308M | 73.5 | 93.1 | 96.4 | 87.0 | 98.4 | 99.5 |
| UNITER$_{BASE}$ (Chen et al., 2020) | 4M | - | 126M | 72.52 | 92.36 | 96.08 | 85.90 | 97.10 | 98.80 |
| UNITER$_{LARGE}$ (Chen et al., 2020) | 4M | - | 343M | 75.56 | 94.08 | 96.76 | 87.30 | 98.00 | 99.20 |
| VILLA$_{LARGE}$ (Gan et al., 2020) | 4M | - | 343M | 76.26 | 94.4 | 96.84 | 87.90 | 97.50 | 98.80 |
| ERNIE-ViL$_{LARGE}$ (Yu et al., 2021) | 4M | 358M | 480M | 76.66 | 94.16 | 96.76 | 89.20 | 98.50 | 99.20 |
| UNIMO$_{LARGE}$ (Li et al., 2021b) | 5.7M | 1.5B | 395M | 78.04 | 94.24 | 97.12 | 89.40 | 98.90 | 99.80 |
| ViLT (Kim et al., 2021) | 4M | 819M | 114M | 64.4 | 88.7 | 93.8 | 83.5 | 96.7 | 98.6 |
| ALBEF(4M) (Li et al., 2021a) | 4M | 120M | 500M | **82.8** | **96.7** | 98.4 | **94.3** | 99.4 | 99.8 |
| METER (Dou et al., 2022b) | 4M | 410M | 384M | 82.22 | 96.34 | 98.36 | 94.30 | **99.60** | **99.90** |
| C3 (our) | 4M | 205M | 384M | 82.64 | 96.52 | **98.42** | 94.21 | 99.50 | 99.80 |

Table 6: Performance comparison of zero-shot image-text retrieval on Flickr30K.

| Model | Images | Iters | Params | Flickr30K-ZS | | | | | |
| | | | | Image Retrieval | | | Text Retrieval | | |
| | | | | R@1 | R@5 | R@10 | R@1 | R@5 | R@10 |
|---|---|---|---|---|---|---|---|---|---|
| CLIP (Radford et al., 2021) | 400M | 12.8B | 151M | 68.7 | 90.6 | 95.2 | 88.0 | 98.7 | 99.4 |
| ALBEF(14M) (Li et al., 2021a) | 14M | 420M | 500M | 82.8 | 96.3 | 98.1 | 94.1 | 99.5 | 99.7 |
| BEIT-3 (Wang et al., 2023) | 36M | 6.1B | 1.9B | 81.5 | 95.6 | 97.8 | 94.9 | 99.9 | 100.0 |
| UNITER$_{BASE}$ (Chen et al., 2020) | 4M | - | 126M | 66.16 | 88.40 | 92.94 | 80.70 | 95.70 | 98.00 |
| UNITER$_{LARGE}$ (Chen et al., 2020) | 4M | - | 343M | 68.74 | 89.20 | 93.86 | 83.60 | 95.70 | 97.70 |
| ViLT (Kim et al., 2021) | 4M | 819M | 114M | 55.0 | 82.5 | 89.8 | 73.2 | 93.6 | 96.5 |
| ALBEF(4M) (Li et al., 2021a) | 4M | 120M | 500M | 76.8 | 93.7 | 96.7 | 90.5 | 98.8 | **99.7** |
| METER (Dou et al., 2022b) | 4M | 410M | 384M | **79.60** | 94.96 | 97.28 | **90.90** | 98.30 | 99.50 |
| C3 (our) | 4M | 205M | 384M | 77.80 | **95.34** | **97.72** | 87.70 | **98.90** | 99.50 |

## C  ABLATION STUDY

We provide the details of Flickr30K in Table 8. Moreover, to better understand the behavior of the C3 model, we also conduct the ablation study on the in-domain dataset, COCO, which is used for pre-training. Table 9 demonstrates the results of image-text retrieval on the COCO dataset. The overall model performance is consistent with the results observed in Flickr30K, NLVR$^2$, and SNLI-VE. For in-domain testing, the correlation between pre-training and testing is vital. Therefore, optimizing the cross-modal features might be more effective than optimizing the single-modal features. Nevertheless, our model still achieves the best performance when incorporating all pre-training objectives (row 4), indicating the effectiveness of learning with concepts.

Table 7: Performance comparison on VQAv2.

| Model | Images | Iters | Params | VQAv2 test-dev | test-std |
|---|---|---|---|---|---|
| ALBEF(14M) Li et al. (2021a) | 14M | 420M | 500M | 75.84 | 76.04 |
| SimVLM$_{HUGE}$(1.8M) Wang et al. (2022) | 1.8B | 4.1B | 632M | 80.03 | 80.34 |
| BEIT-3 (Wang et al., 2023) | 36M | 6.1B | 1.9B | 84.19 | 84.03 |
| UNIMO$_{LARGE}$ Li et al. (2021b) | 5.7M | 1.5B | 395M | 75.06 | 75.27 |
| VinVL$_{LARGE}$ Zhang et al. (2021) | 5.7M | 1B | 520M | 76.52 | 76.60 |
| OSCAR$_{LARGE}$ Li et al. (2020) | 4M | 512M | 380M | 73.61 | 73.82 |
| UNITER$_{LARGE}$ Chen et al. (2020) | 4M | - | 343M | 73.82 | 74.02 |
| VILLA$_{LARGE}$ Gan et al. (2020) | 4M | - | 343M | 74.69 | 74.87 |
| ViLT Kim et al. (2021) | 4M | 819M | 114M | 71.26 | - |
| ALBEF(4M) Li et al. (2021a) | 4M | 120M | 500M | 74.54 | 74.70 |
| METER$_{BASE}$ Dou et al. (2022b) | 4M | 410M | 384M | 77.68 | 77.64 |
| C3 (our) | 4M | 205M | 384M | 77.32 | 77.38 |

Table 8: Ablation study of C3 on Flickr30K. The first row is the METER (Dou et al., 2022b). All results are trained for 2.6k steps on 224×224 images with patch size 32. *I/C* refers to constructing negative samples by misaligned images/concepts. *PW* refers to the pairwise formulation for the MCP objective.

| ITM | MLM | MCP | Flickr30K-ZS IR@1 | IR@5 | IR@10 | TR@1 | TR@5 | TR@10 | Δ |
|---|---|---|---|---|---|---|---|---|---|
| | | | n-gram | | | | | | |
| ✓ | ✓ | - | 31.32 | 61.78 | 74.36 | 39.20 | 68.20 | 79.10 | -5.16 |
| ✓ | - | ✓ | 36.18 | 67.30 | 78.18 | 44.88 | 73.00 | 84.60 | -0.13 |
| - | ✓ | ✓ | 0.80 | 2.92 | 5.14 | 2.30 | 6.30 | 10.10 | -59.56 |
| ✓ | ✓ | ✓ | 36.68 | 67.00 | 77.66 | 46.90 | 73.50 | 83.20 | 0.00 |
| ✓ | ✓ | I | 35.04 | 64.76 | 77.14 | 42.10 | 70.60 | 81.90 | -2.23 |
| ✓ | ✓ | C | 36.80 | 66.22 | 77.20 | 43.30 | 72.60 | 82.70 | -1.02 |
| ✓ | ✓ | PW | 34.44 | 65.84 | 76.32 | 41.90 | 72.50 | 82.60 | -1.89 |
| | | | noun phrase | | | | | | |
| ✓ | ✓ | ✓ | 34.10 | 64.78 | 76.62 | 40.40 | 71.80 | 81.40 | -2.64 |

Table 9: Ablation study of C3 on COCO. The first row is the METER (Dou et al., 2022b). All results are trained for 2.6k steps on 224×224 images with patch size 32. *I/C* refers to constructing negative samples by misaligned images/concepts. *PW* refers to the pairwise formulation for the MCP objective.

| ITM | MLM | MCP | COCO-ZS (5k) IR@1 | IR@5 | IR@10 | TR@1 | TR@5 | TR@10 | Δ |
|---|---|---|---|---|---|---|---|---|---|
| | | | n-gram | | | | | | |
| ✓ | ✓ | - | 24.89 | 53.02 | 66.56 | 32.58 | 61.00 | 72.50 | -3.80 |
| ✓ | - | ✓ | 27.91 | 57.05 | 69.87 | 34.06 | 63.36 | 74.96 | -1.02 |
| - | ✓ | ✓ | 0.41 | 1.77 | 3.23 | 0.64 | 3.18 | 5.88 | -53.04 |
| ✓ | ✓ | ✓ | 29.17 | 57.83 | 70.70 | 35.70 | 64.16 | 75.80 | 0.00 |
| ✓ | ✓ | I | 27.56 | 56.50 | 69.47 | 33.08 | 62.30 | 74.52 | -1.65 |
| ✓ | ✓ | C | 29.15 | 57.96 | 70.77 | 34.26 | 63.76 | 75.62 | -0.31 |
| ✓ | ✓ | PW | 27.86 | 56.84 | 69.72 | 33.78 | 63.44 | 75.86 | -0.98 |
| | | | noun phrase | | | | | | |
| ✓ | ✓ | ✓ | 27.32 | 56.02 | 69.18 | 32.72 | 61.64 | 73.56 | -2.15 |

# D    VISUALIZATION

Fig. 5 provides more visualization for C3 and METER. For each image, we provide two concepts for comparison.

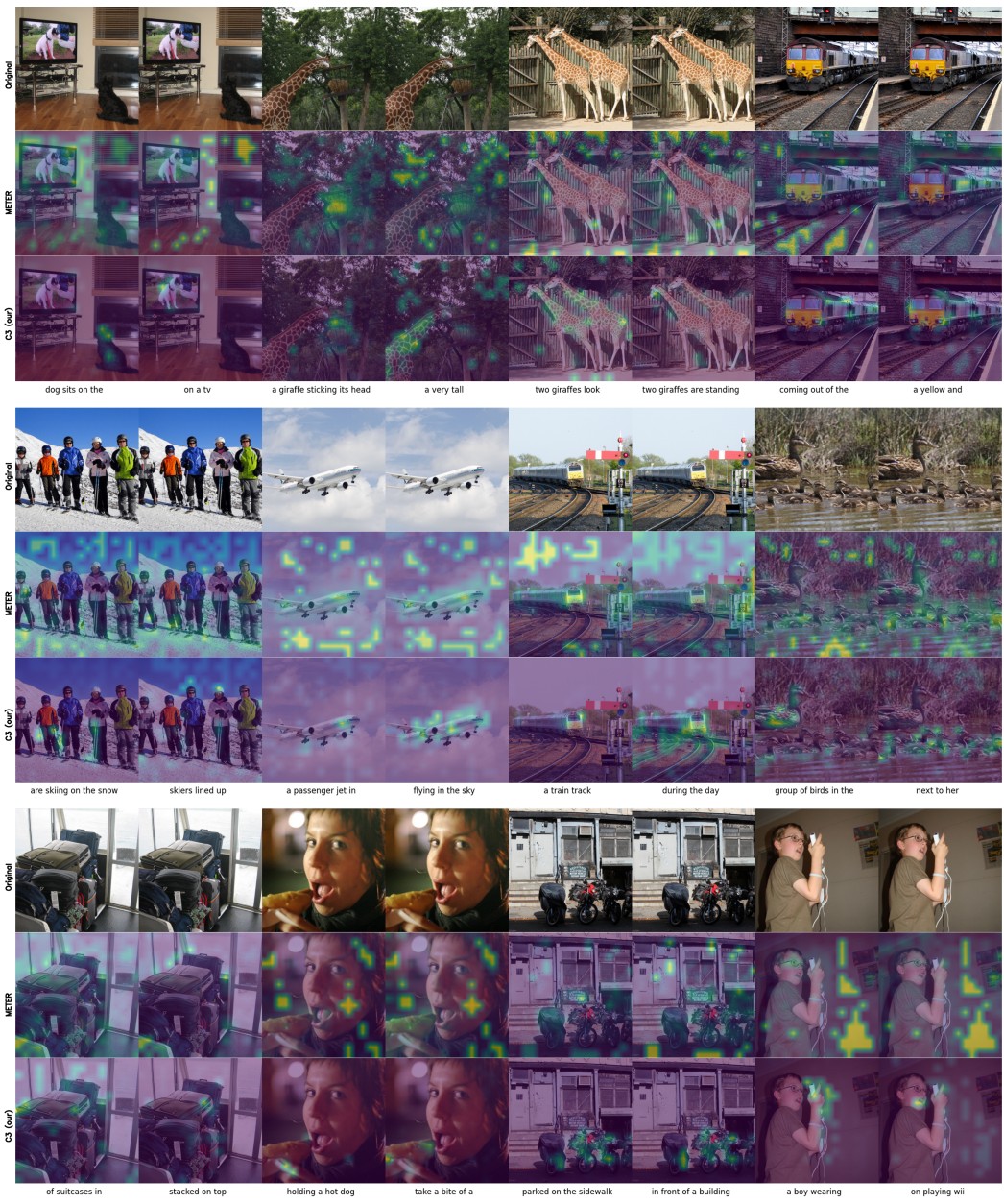

Figure 5: More visualization examples for METER and C3.

# E   CONCEPT DATABASE ANALYSIS

## E.1   EXAMPLES OF CONCEPTS

In this section, we present the findings of our investigations into mined concepts utilizing the proposed $n$-gram strategy and the baseline noun-phrase strategy, which are showcased in Table 10. Our analysis reveals that the $n$-gram approach provides a higher degree of diversity in the captured concepts as compared to the baseline noun-phrase strategy. Specifically, in the first example (row 1), the $n$-gram approach identifies a greater range of concepts, including "to swing a baseball bat" and "while standing on top of," whereas the baseline strategy is more limited. Similarly, in the second example (row 2), the $n$-gram approach identifies the concept of "a grassy field," which is not captured by the baseline approach. The greater diversity of concepts captured through our $n$-gram approach enhances our model's ability to learn concept-level alignments without being confined to pre-defined categories or part-of-speech. These examples underscore the potential of the $n$-gram strategy as a powerful tool in the mining of concepts in the vision-language domain.

Table 10: Concept examples from COCO (Chen et al., 2015). An example includes five captions for each image.

| Image | Captions | Concepts by n-gram mining | Concepts by noun phrase mining |
|---|---|---|---|
|  | (1) An old picture of a baseball player holding a baseball bat. (2) A black and white image depicting a man preparing to swing a baseball bat. (3) A man holding a bat while standing on top of a field. (4) An old fashioned picture shows a baseball batter in uniform. (5)A black and white picture of a baseball player. | black and white picture of, a black and white picture, old picture of a baseball, an old fashioned picture shows, a bat while standing on, holding a bat while standing, a black and white image, picture of a baseball player, to swing a baseball bat, baseball player holding a baseball, player holding a baseball bat, a man preparing to swing, man holding a bat while, man preparing to swing a, a man holding a bat, while standing on top of | a black and white picture, a black and white image, a baseball bat, an old picture, a bat |
|  | (1) In a grassy field is a puppy and a cat who are rubbing noses. (2) A small puppy standing next to a small kitten. (3) The puppy and kitten are in a field of grass. (4) A dog and a cat that are standing in the grass. (5) A kitten is touching noses with a puppy outside. | that are standing in the, standing next to a small, are standing in the grass, in a grassy field, field is a, are in a field of, in a field of grass, a grassy field is, a small kitten, a small puppy, a cat that are, a dog and a cat, and a cat that | a small kitten, a puppy, who, the puppy, noses |
|  | (1) A couple of cars driving through a snow covered street. (2) Vehicle traffic on a city street in a snow storm. (3) Some cars in the street covered with snow. (4) A night time view of a snowy city street. (5) a group of cars that are on some snow | cars that are on some, a night time view of, of cars that are on, a group of cars that, night time view of a, driving through a, some cars in, on a city street in, in a snow storm, vehicle traffic on a, in the street, traffic on a city street, through a snow covered, a city street in a, view of a snowy city, of cars driving through, a snow covered street, group of cars that are, couple of cars driving, cars in the, a couple of cars, the street covered with snow, of a snowy city street, are on some snow | the street, some snow, a snowy city street, a group, a snow storm, a couple, a snow covered street, vehicle traffic, snow, some cars, that, a city street |

## E.2    ABSTRACTION DEGREE FOR CONCEPTS

To gain further insights into the mined concepts in our database, we propose a methodology to quantify their abstraction degree using the attention maps generated by a trained C3 model. Specifically, we define two measurements, namely the *dispersion degree* (DD) and *salient density* (SD), which provide a comprehensive understanding of the abstractness of the concepts. We consider that a concept tends to be more abstract when the model's attention disperses around the visual space, which indicates that the concept is abstract and the model struggles to focus on a specific location. To measure the dispersion tendency, we calculate the entropy of the attention maps. Furthermore, we posit that an abstract concept might involve more salient spots or areas to express the idea. To determine the salient area, we perform connected component analysis [4] on the attention maps, removing the background components (without attention scores) and normalizing the sum of the areas of each attended component to the entire image. The resulting scatter plot of the concepts in terms of DD and SD (Fig. 6a) suggests that the concepts in the top right are more abstract than the ones in the bottom left [5]. We observe that DD and SD have a positive correlation with each other. Additionally, we analyze the correlation between the abstraction degree and text length [6], as shown in the box plots of DD and SD against text length in Fig. 6b. Our findings indicate that DD has a higher correlation with text length than SD. Finally, we provide the visualization examples for three different types of concepts in Fig. 7, which are (a) high DD & high SD, (b) high DD & low SD, and (c) low DD & low SD. These examples correspond to three corners of the scatter plot in Fig. 6a. Interestingly, we notice that the concepts in (c) tend to contain more concrete entities than those in (a) or (b).

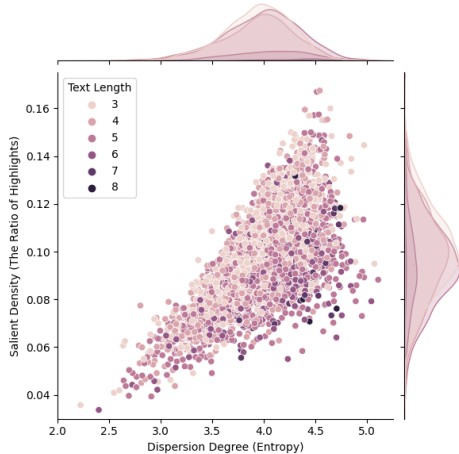

(a) The scatter plot for the dispersion degree and salient density.

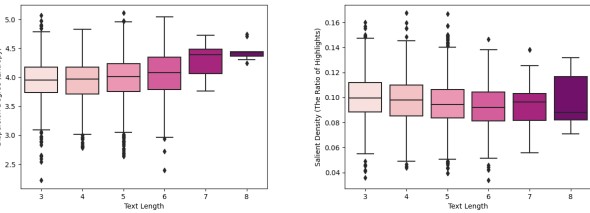

(b) The box plots for the dispersion degree, salient density and text length.

Figure 6: The analysis for the concept abstraction degree in the concept database by the dispersion degree and the salient density.

---

[4]We use *connectedComponentsWithStats()* in the OpenCV to find each component from attention maps, where we set the connectivity to 8.

[5]For each concept, the score of dispersion degree and salient density are averaged over 12 attention heads.

[6]We count the text length after tokenizing the concept text with the tokenizer of https://huggingface.co/facebook/bart-base.

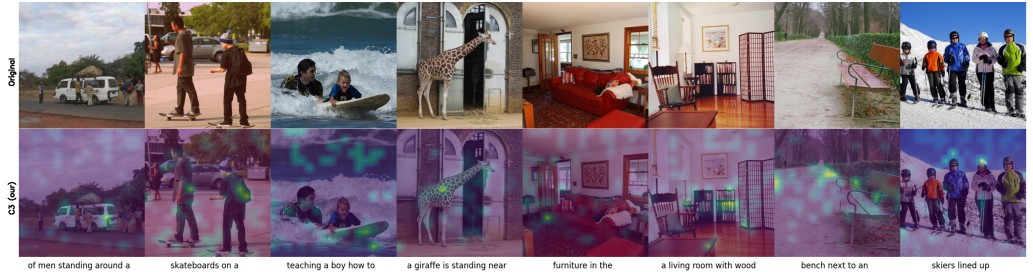

(a) Examples of data with high dispersion degree and high salient density.

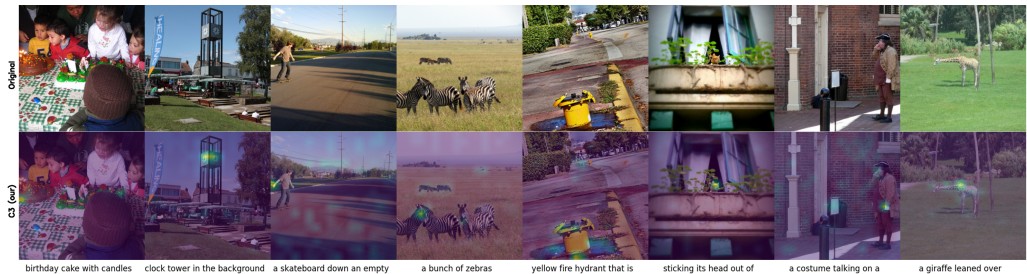

(b) Examples of data with high dispersion degree and low salient density.

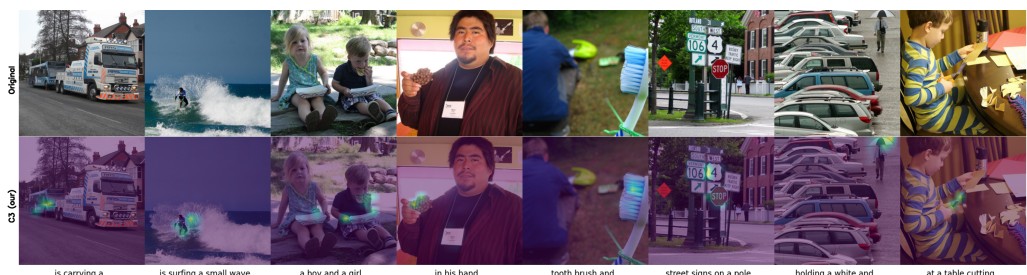

(c) Examples of data with low dispersion degree and low salient density.

Figure 7: Illustration for the concepts with various abstraction degrees defined by dispersion degree and salient density.

Table 11: The zero-shot performance on Flickr30K under different data selection strategies for two models. All models are trained for 2.6k steps on $224 \times 224$ images with patch size 32.

| Model | Data Size | Selection Strategy | Flickr30K-ZS | | | | | | Average |
|-------|-----------|--------------------|--------------|----|-----|----|----|-----|---------|
| | | | Image Retrieval | | | Text Retrieval | | | |
| | | | R@1 | R@5 | R@10 | R@1 | R@5 | R@10 | |
| METER | 567K | - | 31.32 | 61.78 | 74.36 | 39.20 | 68.20 | 79.10 | 58.99 |
| C3 | 567K | - | 37.96 | 67.62 | 78.22 | 46.10 | 73.90 | 83.80 | 64.60 |
| METER | 280K | Random | 28.14 | 58.08 | 69.82 | 36.50 | 65.00 | 75.40 | 55.49 |
| METER | 280K | Low Abstraction Degree | 27.04 | 57.58 | 70.14 | 37.60 | 64.30 | 74.70 | 55.23 |
| METER | 280K | High Abstraction Degree | 31.02 | 61.44 | 73.04 | 40.10 | 67.60 | 78.70 | 58.65 |
| C3 | 280K | Random | 33.60 | 63.26 | 75.48 | 43.50 | 71.90 | 81.60 | 61.56 |
| C3 | 280K | Low Abstraction Degree | 33.90 | 63.72 | 75.44 | 42.30 | 69.80 | 80.30 | 60.91 |
| C3 | 280K | High Abstraction Degree | 37.60 | 67.68 | 77.80 | 45.80 | 74.80 | 83.40 | 64.51 |

In order to understand the impact of text abstraction level on vision-language representation learning, we conduct a deliberate selection of data with different abstraction degrees [7] for pre-training and compared it with a randomly selected dataset. Specifically, we conduct an ablation study using

---

[7]Abstraction degree is calculated based on image captions.

only half of the COCO dataset for pre-training, which is constructed based on data with the top 50% high or low abstraction degree. Our results, as presented in Table 11, reveal that pre-training on data with a high abstraction degree significantly improves data representation quality and even yields competitive performance compared to using the full COCO dataset. Besides, C3 can better increase the benefits compared to METER. These results suggest that the degree of text abstraction is an important factor in vision-language representation learning and that selecting data with a higher abstraction degree can enhance the effectiveness of pre-training.

# F  ANALYSIS OF MASKING RATIO

In this section, we investigate the model behavior under different masking ratios. We utilize the CLIP-ViT-224/16 as our visual encoder. We first pre-train our models using ITM and MLM objectives and evaluate their performance on the zero-shot Flickr30K dataset. Our findings reveal that the commonly used masking ratio of 0.15 may not always be optimal, and Figure 8 demonstrates that the ideal masking ratio ranges from 0.3 to 0.6. Furthermore, we observe that the performance of the C3 model improves at higher masking ratios, as illustrated in Table 12. The results suggest that the choice of masking ratio is crucial for achieving optimal performance across different pre-training settings.

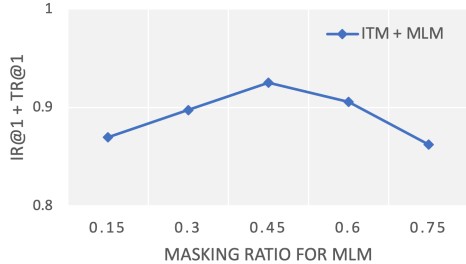

Figure 8: The zero-shot performance on Flickr30K under different masking ratios. We find that increasing the masking ratio to a certain degree improves the performance.

Table 12: The zero-shot performance on Flickr30K under different masking ratios for two models. All models are trained for 2.6k steps on $224 \times 224$ images with patch size 16.

| Model | Masking Ratio | Flickr30K-ZS | | | | | |
|---|---|---|---|---|---|---|---|
| | | Image Retrieval | | | Text Retrieval | | |
| | | R@1 | R@5 | R@10 | R@1 | R@5 | R@10 |
| METER | 0.3 | 38.74 | 70.16 | 80.58 | 51.00 | 78.20 | 85.60 |
| METER | 0.6 | 39.08 | 70.84 | 81.28 | 51.50 | 76.50 | 86.20 |
| C3 | 0.3 | 45.10 | 76.02 | 85.38 | 56.60 | 81.40 | 90.10 |
| C3 | 0.6 | 46.92 | 76.76 | 85.66 | 57.70 | 82.80 | 90.20 |

