# OpenReview forum: "Capture Concept through Comparison: Vision-and-Language Representation Learning with Intrinsic Information Mining"
_ICLR.cc/2024/Conference — Submitted to ICLR 2024_

### Official Review · Reviewer_xrjY · 2023-10-29

**Soundness:** 3 good
**Presentation:** 3 good
**Contribution:** 2 fair
**Rating:** 6
**Confidence:** 3

**Summary:**

This paper proposes the C3 approach, or "Capture Concept through Comparison", which is a multi-modal pretraining framework focusing on aligning abstract concepts between the visual and language modalities.

Specifically, the framework consists of two main components: a concept mining procedure, and a set of training objectives. During concept mining, concepts are extracted as n-grams from text annotations of images. This results in an augmented image-text dataset where each image is accompanied by not only a text description, but a set of "concepts" (n-grams). The augmented dataset is used to pretrain the model using two objectives, namely the Matched Concept Prediction (MCP) loss and the Matched Concept Contrastive (MCC) loss. The MCP loss aims at predicting whether a concept C is shared between two images. The MCC loss encourages the concept-conditioned visual features of two images to be close, should they share the same concept.

The proposed C3 framework is assessed on four image-caption datasets, under the continual pretraining and pretraining-from-scratch configurations, with various widely used pair-centric objectives. Evaluation using the VL-Checklist benchmark shows that model trained with C3 achieves improved performance. An ablation study and a visualisation are provided.

**Strengths:**

- The proposed C3 framework is conceptually correct, in the way that a concept can be better emphasised by comparing images sharing that concept.
- The extensive experimental results clearly showed its efficacy under different configurations for visual-language pretraining.
- The appendix provides an interesting study on the abstraction degree of concepts, using the "dispersion degree" and the "salient density" metrics.

**Weaknesses:**

- The intuition of this paper, which is existing methods only focus on tangible concepts and are therefore less effective when processing abstract concepts like "side by side", is not well reflected by the proposed method. Mining n-grams does make it possible to cluster images with a certain concept, however, how does this reinforce learning of abstract concepts?
- The paper lacks details on how n-grams are collected and processed. For example, it is apparent that not all n-grams are meaningful. How does the framework select useful n-grams?
- The paper claims that C3 achieves better results than baseline models with fewer iterations. While the count of iterations has been reduced, the training using C3 involves an augmented triplet dataset, where the n-grams serve as extra signals for comparison between images. Therefore, C3's computation burden per iteration could be much higher than the compared methods. It is therefore preferable to compare the wall time or FLOPS per epoch if the paper is trying to highlight the efficiency.
- The visualisation provided in Figure 4 is not convincing enough. For example, Figure 4(c) shows "looking at a bird", why is the bird not highlighted? Figure 4(g) is about "ready to hit a ball", but why are hands the only highlighted region while this preparation posture obviously involves legs? The interpretation of concepts can be highly subjective and therefore I suggest the paper provides some quantitative studies about this. For example, it can provide the classification of model on a subset of abstract concepts trained with and without C3.

**Questions:**

Please see the weakness points above. As a summary here, I expect some answers for these questions:
1. How does the n-gram based learning reinforce modelling of abstract concepts?
2. How are n-grams selected? Can the authors provide more statistics about the collected n-grams?
3. How much time does C3 take to train a model? Alternatively, how much computation (measured by, for example, FLOPS) is needed? Is this still less than the baseline models?
4. Can the authors provide quantitative evidence that modelling of abstract concepts is indeed improved by C3?

---

> ### Author Response · Authors · 2023-11-18
> **Reply from Authors**
>
> 1. **The intuition of this paper, which is existing methods only focus on tangible concepts and are therefore less effective when processing abstract concepts like "side by side", is not well reflected by the proposed method. Mining n-grams does make it possible to cluster images with a certain concept, however, how does this reinforce learning of abstract concepts?**
>
> Thank you for your insightful question. In our approach, the n-gram mining procedure is deliberately designed to be inclusive of all word properties, thus facilitating the extraction of a diverse array of concepts, both concrete and abstract, from real-world data. This inclusive strategy is pivotal in capturing a comprehensive spectrum of concepts embedded in sentences, which is essential for learning nuanced semantics. This, in turn, significantly enhances vision-language understanding, especially in the context of abstract concepts that have been comparatively underserved by existing methods.
>
> We consciously avoid implementing a filter specifically for abstract concepts during the learning process. This decision stems from the ongoing debate and lack of consensus in the academic community about what precisely constitutes 'abstractness.' Implementing such a filter prematurely could risk narrowing the concept diversity in our dataset, which might adversely affect the overall learning process and performance. We assert that the rich tapestry of concrete and abstract concepts together enriches the learning process, providing a more holistic understanding of language in its myriad forms. Therefore, our method, through its inclusive approach, indirectly but effectively bolsters the understanding of abstract concepts, addressing the gap identified in existing methods.
>
> 2. **The paper lacks details on how n-grams are collected and processed. For example, it is apparent that not all n-grams are meaningful. How does the framework select useful n-grams?**
>
> Thank you for your comment. In our approach, we apply a predetermined frequency threshold to filter n-grams. This ensures that the concepts we select are those that appear in real-world data with a frequency exceeding a certain predefined level.

---

> ### Author Response · Authors · 2023-11-18
> **Reply from Authors (Continue)**
>
> 3. **The paper claims that C3 achieves better results than baseline models with fewer iterations. While the count of iterations has been reduced, the training using C3 involves an augmented triplet dataset, where the n-grams serve as extra signals for comparison between images. Therefore, C3's computation burden per iteration could be much higher than the compared methods. It is therefore preferable to compare the wall time or FLOPS per epoch if the paper is trying to highlight the efficiency.**
>
> Thank you for bringing this question. We appreciate the opportunity to provide additional analysis and clarify the rationale behind our training decisions.
>
> The use of triplets does indeed incur additional computational costs; however, the training costs introduced by triplets are lower compared to the combined costs of ITM (Image Text Matching) and MLM (Masked Language Modeling). Consequently, after halving the iterations, the overall training time of C3 is still lower than that of METER.
>
> In response to the reviewer's feedback, we conducted additional supplementary experiments on learning efficiency. We first provide the image and text usage and the corresponding computational cost for each objective and compare the training wall time per epoch, as shown in the following table. METER adopts ITM and MLM objectives, while the learning of C3 further combines MCP or MCC. Furthermore, to align the data usage and computational cost per training step for METER and C3, we extended METER by augmenting the ITM and MLM with extra image-text pairs, denoted as METER+. This data expansion ensures comparable computational complexity under the same training step, with METER+ slightly exceeding C3 due to an extra caption encoding process. To showcase the learning efficiency of C3, we present the pre-training wall time over the performance of the zero-shot retrieval task in the following figure (https://anonymous.4open.science/r/Anonymous-2514/README.md). The results consistently indicate C3’s superiority over baselines within the same time frame. Please note the training time includes the data processing and thus may not precisely reflect the theoretical cost.
>
> ---- | Image usage | Text usage | Computational cost per step | Wall time per epoch
> ----|----|----|----|----
> ITM / MLM |  $v$ | $t$ | $E_v+E_t+E_{cross}$ | -
> MCP / MCC | $v$+$v_m$ | $c$ | $2E_v+E_t+2E_{cross}$ | -
> METER  | $2v$ | $2t$ |	$2E_v+2E_t+2E_{cross}$ | 0.52 hr
> METER+ | $3v+v_m$| $3t+t_m$  | $4E_v+4E_t+4E_{cross}$ | 0.67 hr
> C3 | $3v+v_m$| $2t+c$ | $4E_v+3E_t+4E_{cross}$ | 0.63 hr
>
> Table: Data usage and computational cost per step for objectives and models. $v$ and $v_m$ denote the image and matched image, while $t$ and $t_m$ denote the caption of  $v$ and $v_m$. $c$ is the concept text. Note: In practice, the image  $v$ is the same for objectives.
>
> 4. **The visualisation provided in Figure 4 is not convincing enough. For example, Figure 4(c) shows "looking at a bird", why is the bird not highlighted? Figure 4(g) is about "ready to hit a ball", but why are hands the only highlighted region while this preparation posture obviously involves legs? The interpretation of concepts can be highly subjective and therefore I suggest the paper provides some quantitative studies about this. For example, it can provide the classification of model on a subset of abstract concepts trained with and without C3.**
>
> Thank you for your feedback. The visualization could provide a qualitative evaluation to help us investigate the behavior of models. Furthermore, our abstractness experiments in Appendix E.2 function as a means to quantify the utility of model attentions. In Appendix E.2, we introduce two metrics—dispersion degree and salient density—derived from attention maps to measure the data abstractness. This analysis, conducted on COCO pre-training datasets, involves filtering the top 50% abstractness data for pre-training. Table in Appendix E.2 displays the corresponding downstream performance, revealing that training with the selected data nearly restores performance comparable to full re-training. Implicitly, these experiments suggest that attentions from C3 effectively capture features that are more informative or challenging, thereby enhancing model learning efficiency. This approach offers a method to quantify attention properties beyond human judgment.

---

### Official Review · Reviewer_J1vN · 2023-10-30

**Soundness:** 3 good
**Presentation:** 3 good
**Contribution:** 3 good
**Rating:** 5
**Confidence:** 4

**Summary:**

The paper introduces a concept-centric pre-training method for improving the text-image alignment properties of vision-language models. It introduces a data mining formulation using n-grams to find image-text pairs containing text with similar concepts and uses this to formulate loss functions for image-text alignment. The framework is primarily evaluated on the VL-Checklist, where it shows quantitative gains over the baselines, and is also evaluated on 2 other datasets where it performs competitively with other baselines.

**Strengths:**

+ The paper introduces concept-based pre-training, which is an interesting way of improving the alignment of VLMs.
+ The idea is simple, intuitive, and easy to follow.
+ The evaluation of the image-text alignment is performed on the VL-Checklist, which is a standardized benchmark for the task.
+ The quantitative gains over considered baselines is clear on the VL-Checklist, which shows the effectiveness of the approach.

**Weaknesses:**

- The main contribution is the concept mining framework and the resulting MCP/MCC loss functions. While the ablation study focuses on the effect of not using the loss functions, there is no evaluation of the actual triplets mined. Some qualitative examples are given in the supplementary, but there is no evaluation of the effect of this mining framework. For example, K1 and K2 are set to 5 and 80 (page 6, paragraph 5)). Why these numbers? What is their significance? How were they selected?
- While the experimental results show improvement, the considered tasks are still somewhat related, i.e., falling within the same umbrella as text-image matching. Does the pre-training/finetuning strategy improve the performance on downstream tasks such as VQA, zero-shot classification, etc.? It would be good to show additional gains on tasks other than the ITM task.
- The gains over METER, the main source of comparison, seem to be limited on tasks other than the ones in the VL-Checklist. Is there any reason why? Not much qualitative discussion is present, so it is hard to understand why the gains do not transfer to other tasks.
- Why n-grams and not semantic similarity between captions/sentences/noun phrases? It is an ablation study that can show the impact of choices in the framework, considering the concept mining paradigm is the core contribution.
- Algorithm 1 seems incomplete. What is C_k? I assume it is an empty set like I_k. Need to show the role of K1 and K2 in the algorithm. K1 and K2 are mentioned in passing in the text without detail. This would be a good place to introduce it.
- Minor: writing could be improved by denoting what the abbreviations are when they are introduced. For example, the term VLP is thrown in Section 3 without any prior references to this. Given the many acronyms in the vision/ML community, expanding it would help readability. For the record, I think it refers to vision-language pre-training, but I am not entirely sure.

**Questions:**

My major concerns are based on the ablation studies and the choice of evaluation benchmarks beyond image-text matching. There is no evidence of generalization beyond the ITM tasks that somewhat limits its contributions. Please see the weaknesses section for more details.

---

> ### Author Response · Authors · 2023-11-18
> **Reply from Authors**
>
> 1. **The main contribution is the concept mining framework and the resulting MCP/MCC loss functions. While the ablation study focuses on the effect of not using the loss functions, there is no evaluation of the actual triplets mined. Some qualitative examples are given in the supplementary, but there is no evaluation of the effect of this mining framework. For example, K1 and K2 are set to 5 and 80 (page 6, paragraph 5)). Why these numbers? What is their significance? How were they selected?**
>
> Thank you for your comment. We conducted an ablation study on our concept mining by confining concepts to noun phrases, as indicated in the last row of Table 3. The results confirm that our n-gram mining procedures could better enhance the model's capability by yielding more diverse and informative concepts without specific constraints.
>
> The selection of hyperparameters K_1 and K_2 is guided by pre-training dataset statistics. Given the inherent long-tail distribution of concepts, our objective is to balance the amount of various concepts, preventing the dominance of commonly observed ones. K_1 is introduced to limit the number of retrieved images per concept in sentences, empirically set at 5. Note that a concept may appear in different sentences, so the retrieved image count for a concept is constrained by (number of occurrences * 5). Concerning K_2, our statistics indicate that the majority of data in the pre-training dataset involves approximately 14-16 concepts. Accordingly, we set K_2 at 80 (i.e., K_1 * 16) as the upper limit for the number of retrieved images per data point. These values primarily stem from dataset statistics. Notably, we did not conduct extensive experiments on the hyperparameters, so there may exist other settings that can further optimize the model performance.
>
> 2. **While the experimental results show improvement, the considered tasks are still somewhat related, i.e., falling within the same umbrella as text-image matching. Does the pre-training/finetuning strategy improve the performance on downstream tasks such as VQA, zero-shot classification, etc.? It would be good to show additional gains on tasks other than the ITM task.**
>
> Thank you for your suggestions. We have conducted experiments on VQAv2 and present the results below.
>
> Model | test-dev | test-std
> ----|----|----
> ViLT (ICML 2021) | 71.26 | -
> VILLA (NeuIPS 2020) | 74.69 | 74.87
> ALBEF (NeuIPS 2021) | 74.54 | 74.70
> METER (CVPR 2022) | 77.68 | 77.64
> C3 | 77.32 | 77.38
>
> The findings reveal that C3 performs on par with METER, which is our backbone model. These similar results can be attributed to the comparatively lower demands for abstract or complex concept understanding in VQAv2 questions. To substantiate this claim, we aim to quantify the text abstractness for non-retrieval tasks, including VQAv2, SNLI-VE and NLVR2. Specifically, We utilize the SemCat dataset [R1], which comprises approximately 6,500 English words. Each word in this dataset is annotated for abstractness by humans on a scale from 1 to 5, as detailed in references [R2, R3]. We define sentence-level abstractness as the average abstractness of the top 30% of words. Besides, we also adapt the metrics we proposed in Section 4.5 to quantify the semantic complexity. The results are shown as follows:
>
> Dataset | Abstractness | Char. Num | Token Num | Tree Depth
> ----|----|----|----|----
> VQAv2 | 2.77 +- 0.28 | 31.07 | 7.29  | 4.73
> SNLI-VE | 2.96 +- 0.33 | 37.72   | 8.30 | 5.63
> NLVR2 | 3.36 +- 0.27 |  72.42  | 15.54 |  7.33
>
> The results indicate that VQAv2 exhibits significantly lower abstractness compared to other benchmarks, supporting our hypothesis that VQAv2 places less emphasis on understanding abstract concepts. In a broader context, we consider that diverse datasets may necessitate varying degrees of abstract concept comprehension. Nonetheless, our C3 model can still achieve comparable results with the backbone models in cases where a high level of abstractness understanding is not highly demanded. It is crucial to emphasize that our ultimate goal is to enhance the overall foundational capabilities of VLP models, as demonstrated by our focus on reporting results on the VL-checklist.
>
> [R1] Semantic Structure and Interpretability of Word Embeddings, TASLP, 2018. \
> [R2] Explainable Semantic Space by Grounding Language to Vision with Cross-Modal Contrastive Learning, NeuIPS, 2021. \
> [R3] Concreteness ratings for 40 thousand generally known English word lemmas, Behavior Research Methods, 2014

---

> ### Author Response · Authors · 2023-11-18
> **Reply from Authors (Continue)**
>
> 3. **The gains over METER, the main source of comparison, seem to be limited on tasks other than the ones in the VL-Checklist. Is there any reason why? Not much qualitative discussion is present, so it is hard to understand why the gains do not transfer to other tasks.**
>
> Thank you for your comment. Our ultimate target is to enhance the model's proficiency in understanding abstract concepts. However, the extent to which tasks necessitate this ability varies, leading to divergent benefits from our methods. To illustrate,  in Section 4.5, we conducted experiments on different splits of NLVR2, specifically selecting data based on complexity. The results indicate that our methods yield greater performance gains on more abstract and complex data.
>
> While recognizing that the advantages of improved abstract concept understanding may not be universally applicable across tasks, clear enhancements are evident in datasets that strongly prioritize this ability, such as VL-Checklist and tailored splits for NLVR2.
>
> 4. **Why n-grams and not semantic similarity between captions/sentences/noun phrases? It is an ablation study that can show the impact of choices in the framework, considering the concept mining paradigm is the core contribution.**
>
> Thank you for your comment. We begin by revisiting the n-gram mining process for clarity. Given an <image_1, text_1> pair, we extract n-grams from text_1 and utilize them to retrieve additional pairs, e.g., <image_2, text_2>, where text_1 and text_2 share an overlapping n-gram, referred to as a concept. Subsequently, we formulate training examples as <image_1, image_2, concept>.
>
> We justify the utilization of n-gram similarity for two primary reasons:
> - N-gram matching operates universally at the text level, ensuring high generalizability across domains. This mitigates reliance on trained models for creating vectorized representations, which can vary and introduce biases from the learned data.
> - The extraction of features for text fragments, specifically concepts in our methodology, lacks a well-established method. While approaches like simple averaging of word embeddings are conceivable, past research [R1] indicates potential limitations in providing salient semantic information. Additionally, our observations reveal a lower frequency of abstract concepts in real-world data, raising concerns about the quality of features for such concepts.
>
> Hence, we advocate leveraging n-gram similarity in our mining procedure.
>
> [R1] Sentence-BERT: Sentence Embeddings using Siamese BERT-Networks, EMNLP, 2019
>
> 5. **Algorithm 1 seems incomplete. What is C_k? I assume it is an empty set like I_k. Need to show the role of K1 and K2 in the algorithm. K1 and K2 are mentioned in passing in the text without detail. This would be a good place to introduce it.**
>
> Thank you for your comment. We apologize for the ambiguity. The $\mathcal{C}_k$ (in calligraphy font) in the algorithm represents the k-th concept of a text and is not a set. To enhance clarity, we have revised it to $C_k$ (in italic font). Additionally, we have incorporated $K_1$ and $K_2$ into the algorithm in the revised paper.
>
> 6. **Minor: writing could be improved by denoting what the abbreviations are when they are introduced. For example, the term VLP is thrown in Section 3 without any prior references to this. Given the many acronyms in the vision/ML community, expanding it would help readability. For the record, I think it refers to vision-language pre-training, but I am not entirely sure.**
>
> Thanks for your suggestion. We have modified the descriptions in Section 3 for better clarity.

---

### Official Review · Reviewer_snHQ · 2023-10-31

**Soundness:** 3 good
**Presentation:** 4 excellent
**Contribution:** 3 good
**Rating:** 6
**Confidence:** 4

**Summary:**

The paper presents 3 main contributions -- (a) a data mining procedure to create <image, image, text> triplets such that the two images are conceptually related to the text, (b) concept centric learning objectives created for the training dataset containing the mined triplets, (c) empirical results to show the benefits of (a) and (b).

The empirical results are shown under the settings of pretraining and parameter-efficient finetuning using LoRA. Additionally, ablation studies are also included to study various aspects of the learning objectives.

**Strengths:**

* The paper is well-written and organized.

* The data mining method is graceful and can be scaled to other domains / large-scale datasets.

* The concept centric pretraining objectives -- MCP and MCC -- to maximize the mutual information between the training triplets are intuitive and easy to implement.

* The results in Table 1 are impressive -- especially continual pre-training results on COCO and pretraining from scratch.

**Weaknesses:**

* It is unclear why the authors choose to train C3 in Table 2 for half the number of iterations? Are the other hparams (learning rate, batch size) the same between METER and C3? Is the number of iters halved to account for using two images (due to <image, image, text>) in a single example in C3 compared to one image in METER?

* Also, the improvements in Table 2 for C3 compared to METER seem very small, especially so on test splits of SNLI-VE and NLVR.

* The ablation study in Table 3 does not show very clear patterns, e.g., (a) row 2 for Flickr30k-ZS seems comparable to row 4, (b) all the rows in SNLI-VE seem very close

* Unclear what to make of the visualizations in 4.6 as there is a risk of cherry-picking. Is there a better way to quantify the claim "C3 exhibits the ability to focus on specific regions in accordance with the text fragments"?

**Questions:**

* Is it possible to show an ablation study where <image, image, text> triplets are constructed using an alternate data mining strategy. E.g., instead of looking for n-gram match, create triplets {<image, image, text_1>, <image, image, text_2>} where (text_1, text_2) are semantically similar (but do not necessarily have n-gram overlap).

---

> ### Author Response · Authors · 2023-11-18
> **Reply from Authors**
>
> 1. **It is unclear why the authors choose to train C3 in Table 2 for half the number of iterations? Are the other hparams (learning rate, batch size) the same between METER and C3? Is the number of iters halved to account for using two images (due to <image, image, text>) in a single example in C3 compared to one image in METER?**
>
> Thank you for your thoughtful inquiry. We appreciate the opportunity to clarify the rationale behind our training decisions.
>
> Training C3 for half the number of iterations was a strategic choice aimed at ensuring a fair and comparable evaluation with METER. Given that the MCP and MCC objectives involve incorporating an additional image and concept text into the training procedure, it naturally incurs an extra cost in terms of image and text encoding time. By halving the training iterations, we aimed to balance this additional computational load and make the overall computational cost of C3 even less than that of METER. We maintained uniformity in other hyperparameters, including learning rate and batch size, across both models to isolate the impact of the modified training procedure.
>
> Additionally, we conducted another analysis to understand the learning efficiency of C3 under the same data usage. The following table first provides the image and text usage and the corresponding computational cost for using these objectives. METER adopts ITM and MLM objectives, while the learning of C3 further combines MCP or MCC objectives depending on the backbone model. To align the data usage and computational cost per training step for METER and C3, we extend METER by augmenting the ITM and MLM with extra image-text pairs, denoted as METER+. This data expansion ensures comparable computational complexity under the same training step, with METER+ slightly exceeding C3 due to an extra caption encoding process. To showcase the learning efficiency of C3, we present the pretraining wall time over the performance of the zero-shot retrieval task in the following figure (https://anonymous.4open.science/r/Anonymous-2514/README.md). The results consistently indicate C3’s superiority over baselines within the same time frame. Please note the training time includes the data processing and thus may not precisely reflect the theoretical cost.
>
> ---- | Image usage | Text usage | Computational cost per step | Wall time per epoch
> ----|----|----|----|----
> ITM / MLM |  $v$ | $t$ | $E_v+E_t+E_{cross}$ | -
> MCP / MCC | $v$+$v_m$ | $c$ | $2E_v+E_t+2E_{cross}$ | -
> METER  | $2v$ | $2t$ |    $2E_v+2E_t+2E_{cross}$ | 0.52 hr
> METER+ | $3v+v_m$| $3t+t_m$  | $4E_v+4E_t+4E_{cross}$ | 0.67 hr
> C3 | $3v+v_m$| $2t+c$ | $4E_v+3E_t+4E_{cross}$ | 0.63 hr
>
> Table: Data usage and computational cost per step for objectives and models. $v$ and $v_m$ denote the image and matched image, while $t$ and $t_m$ denote the caption of  $v$ and $v_m$. $c$ is the concept text. Note: In practice, the image  $v$ is the same for objectives.
>
> 2. **Also, the improvements in Table 2 for C3 compared to METER seem very small, especially so on test splits of SNLI-VE and NLVR.**
>
> Thank you for your comment. Our ultimate target is to enhance the model's proficiency in understanding abstract concepts. However, the extent to which tasks necessitate this ability varies, leading to divergent benefits from our methods. To illustrate,  in Section 4.5, we conducted experiments on different splits of NLVR2, specifically selecting data based on complexity. The results indicate that our methods yield greater performance gains on more abstract and complex data.
>
> While recognizing that the advantages of improved abstract concept understanding may not be universally applicable across tasks, clear enhancements are evident in datasets that strongly prioritize this ability, such as VL-Checklist and tailored splits for NLVR2.
>
> 3. **The ablation study in Table 3 does not show very clear patterns, e.g., (a) row 2 for Flickr30k-ZS seems comparable to row 4, (b) all the rows in SNLI-VE seem very close**
>
> Thank you for your comment. We consider the observed inconsistent pattern to be attributed to task-specific properties. Notably, for Flickr30k-ZS, row 2 and 4 indicate its relative insensitivity to MLM objectives, whereas row 3 and row 4 prove that ITM is crucial. Conversely, SNLI-VE exhibits low ITM dependence, with MLM exerting more influence. These findings underscore that distinct pre-training objectives yield varied improvements in downstream tasks, where ITM generally provides sentence-level alignments, and MLM assists in word-level alignments. Moreover, tasks exhibit diverse sensitivities to MCP objectives, yet MCP consistently enhances performance across different tasks (row 1 and row 4).

---

> ### Author Response · Authors · 2023-11-18
> **Reply from Authors (Continue)**
>
> 4. **Unclear what to make of the visualizations in 4.6 as there is a risk of cherry-picking. Is there a better way to quantify the claim "C3 exhibits the ability to focus on specific regions in accordance with the text fragments"?**
>
> Thank you for your feedback. For more additional illustrative examples, please refer to Appendix D in the supplementary materials. To select visualization examples, we filter data with a minimum of two concepts, each consisting of at least three tokens. Subsequently, we randomly sample from the filtered set. We posit that such data elucidates our model's design objectives.
>
> Furthermore, our abstractness experiments in Appendix E.2 function as a means to quantify the utility of model attentions. In Appendix E.2, we introduce two metrics—dispersion degree and salient density—derived from attention maps to measure the abstraction degree. This analysis, conducted on COCO pre-training datasets, involves filtering the top 50% abstractness data for pre-training. Table in Appendix E.2 displays the corresponding downstream performance, revealing that training with the selected data nearly restores performance comparable to full re-training. Implicitly, these experiments suggest that attentions from C3 effectively capture features that are more informative or challenging, thereby enhancing model learning efficiency. This approach offers a method to quantify attention properties beyond human judgment.
>
> 5. **Is it possible to show an ablation study where <image, image, text> triplets are constructed using an alternate data mining strategy. E.g., instead of looking for n-gram match, create triplets {<image, image, text_1>, <image, image, text_2>} where (text_1, text_2) are semantically similar (but do not necessarily have n-gram overlap).**
>
> Thank you for your comment. We begin by revisiting the n-gram mining process for clarity. Given an <image_1, text_1> pair, we extract n-grams from text_1 and utilize them to retrieve additional pairs, e.g., <image_2, text_2>, where text_1 and text_2 share an overlapping n-gram, referred to as a concept. Subsequently, we formulate training examples as <image_1, image_2, concept>.
>
> We justify the utilization of n-gram similarity for two primary reasons:
> - N-gram matching operates universally at the text level, ensuring high generalizability across domains. This mitigates reliance on trained models for creating vectorized representations, which can vary and introduce biases from the learned data.
> - The extraction of features for text fragments, specifically concepts in our methodology, lacks a well-established method. While approaches like simple averaging of word embeddings are conceivable, past research [R1] indicates potential limitations in providing salient semantic information. Additionally, our observations reveal a lower frequency of abstract concepts in real-world data, raising concerns about the quality of features for such concepts.
>
> Hence, we advocate leveraging n-gram similarity in our mining procedure.
>
> [R1] Sentence-BERT: Sentence Embeddings using Siamese BERT-Networks, EMNLP, 2019

---

### Official Review · Reviewer_pYtZ · 2023-10-31

**Soundness:** 3 good
**Presentation:** 3 good
**Contribution:** 3 good
**Rating:** 6
**Confidence:** 3

**Summary:**

Aligning the semantics of images and text is a challenging task. The conventional approaches have tried to improve alignment by adding extra information (tags, bounding boxes) as linkages between the two modalities. However, the existing methods mainly focus on aligning concrete objects and ignore other crucial abstract concepts that are hard to see (e.g., "side by side" and "upside down").

To address this limitation, the paper proposes a new method called "Capture various Concepts through data Comparison (C3)" for learning cross-modal representations. C3 incorporates a novel n-gram-based mining procedure to discover the concepts intrinsic to the database. It also frames the model inputs as triplets (image-image-concept) to capture abstract semantics in images better.

Based on this setup, the paper proposes two concept-centric pre-training objectives, Matched Concept Prediction (MCP) and Matched Concept Contrastive (MCC), to learn concept-level alignment better. Extensive experiments show that models trained with C3 consistently improve performance on various comprehension and reasoning benchmarks, whether starting from scratch or fine-tuning from an existing model.

**Strengths:**

The paper is well-written and easy to understand, particularly with the help of Figure 2 and Algorithm 1, which clearly elucidate the core of the proposed mining strategy. The benefits of the proposed method are clearly explained and demonstrated through a variety of rigorous experiments and evaluations. The experimental results show consistent performance improvement. It reveals that the proposed mining strategy is simple but effective, and it has the potential to be leveraged in other studies well.

**Weaknesses:**

1. The paper has omitted a significant benchmark from the evaluation. VQAv2 is a widely used and crucial benchmark in VLM research, and its performance should be evaluated.

2. With the overload of VLM studies, it is impractical to compare and validate all of them. However, the performance of some well-known methodologies may be added to the results section. BEIT3, for example, is a known high-performing model with a parameter size of about 1.9B, making it a suitable comparison.

3. While the volume of data and the number of iterations in the training process are essential, so too is the parameter size of the model. Adding the parameter size to Table 2 would facilitate the evaluation of the proposed method. C3, for instance, has a parameter size of 400M, which is relatively lightweight. It could be another advantage of the proposed method.

**Questions:**

1. It is difficult to find the ablation result with or without the MCC objective. Is there any difficulty in the ablation process?

---

> ### Author Response · Authors · 2023-11-18
> **Reply from Authors**
>
> 1. **The paper has omitted a significant benchmark from the evaluation. VQAv2 is a widely used and crucial benchmark in VLM research, and its performance should be evaluated.**
>
> Thank you for your suggestions. We have conducted experiments on VQAv2 and present the results below.
>
> Model | test-dev | test-std
> ----|----|----
> ViLT (ICML 2021) | 71.26 | -
> VILLA (NeuIPS 2020) | 74.69 | 74.87
> ALBEF (NeuIPS 2021) | 74.54 | 74.70
> METER (CVPR 2022) | 77.68 | 77.64
> C3 | 77.32 | 77.38
>
> The findings reveal that C3 performs on par with METER, which is our backbone model. These similar results can be attributed to the comparatively lower demands for abstract or complex concept understanding in VQAv2 questions. To substantiate this claim, we aim to quantify the text abstractness for non-retrieval tasks, including VQAv2, SNLI-VE and NLVR2. Specifically, We utilize the SemCat dataset [R1], which comprises approximately 6,500 English words. Each word in this dataset is annotated for abstractness by humans on a scale from 1 to 5, as detailed in references [R2, R3]. We define sentence-level abstractness as the average abstractness of the top 30% of words. Besides, we also adapt the metrics we proposed in Section 4.5 to quantify the semantic complexity. The results are shown as follows:
>
> Dataset | Abstractness | Char. Num | Token Num | Tree Depth
> ----|----|----|----|----
> VQAv2 | 2.77 +- 0.28 | 31.07 | 7.29  | 4.73
> SNLI-VE | 2.96 +- 0.33 | 37.72   | 8.30 | 5.63
> NLVR2 | 3.36 +- 0.27 |  72.42  | 15.54 |  7.33
>
> The results indicate that VQAv2 exhibits significantly lower abstractness compared to other benchmarks, supporting our hypothesis that VQAv2 places less emphasis on understanding abstract concepts. In a broader context, we consider that diverse datasets may necessitate varying degrees of abstract concept comprehension. Nonetheless, our C3 model can still achieve comparable results with the backbone models in cases where a high level of abstractness understanding is not highly demanded. It is crucial to emphasize that our ultimate goal is to enhance the overall foundational capabilities of VLP models, as demonstrated by our focus on reporting results on the VL-checklist.
>
> [R1] Semantic Structure and Interpretability of Word Embeddings, TASLP, 2018. \
> [R2] Explainable Semantic Space by Grounding Language to Vision with Cross-Modal Contrastive Learning, NeuIPS, 2021. \
> [R3] Concreteness ratings for 40 thousand generally known English word lemmas, Behavior Research Methods, 2014
>
> 2. **With the overload of VLM studies, it is impractical to compare and validate all of them. However, the performance of some well-known methodologies may be added to the results section. BEIT3, for example, is a known high-performing model with a parameter size of about 1.9B, making it a suitable comparison.**
>
> Thank you for your suggestion. Accordingly, we have incorporated additional baselines for comparisons as shown in the revised version. Our observations indicate that these methods exhibit strong performance, particularly benefiting from larger model sizes. Importantly, C3 is a versatile framework capable of generalization to diverse architectures, thereby capitalizing on potential scaling advantages. The comparison with the equivalently scaled backbone, METER, clearly highlights the improvement achieved with our C3 framework.
>
> 3. **While the volume of data and the number of iterations in the training process are essential, so too is the parameter size of the model. Adding the parameter size to Table 2 would facilitate the evaluation of the proposed method. C3, for instance, has a parameter size of 400M, which is relatively lightweight. It could be another advantage of the proposed method.**
>
> Thank you for your suggestions. Accordingly, we have incorporated the model size information into tables in the revised version, illustrating our model's superior performance with fewer trainable parameters. This advantage stems from explicit learning through our concept mining procedure. It is also crucial to underscore the generalizability of our approach, indicating it is applicable to large models for scaling benefits.
>
> 4. **It is difficult to find the ablation result with or without the MCC objective. Is there any difficulty in the ablation process?**
>
> Thank you for your comment. In our experimental settings, MCC is employed for architectures without cross-modal interaction (e.g., CLIP), while MCP is applied to those featuring cross-modal interaction (e.g., METER). The ablation of MCC is therefore implicitly shown in the continual pre-training section of Table 1, where rows 1 and 6 represent baselines pre-trained without the MCC objective. Accordingly, we will enhance the table description in our revision for better clarity.
>
> We have made modifications in the revised version for addressing the comments above.

---

### Meta-Review · Area_Chair_uP6m · 2023-12-08

**Metareview:**

The paper proposes a novel pretraining method for improving the text-image alignment properties of vision-language models by formulating a loss function that uses image-image-concept triplets mined by finding image-text pairs whose texts contain a matching concept (n-grams).

The paper received three weak accepts and a weak reject. While all four reviewers seem to acknowledge the novelty of the work, most reviewers also pointed out the lack of experiments and limited performance gains. Especially, Reviewer pYtZ and J1vN both raised a concern about the missing experiments on a VQA benchmark. Although the authors provided additional experimental results, they are clearly weaker than the results in the limited setting in the manuscript.

Overall, the AC recommends improving the paper further by reflecting the comments from the reviewers and submitting it to another venue.

**Justification For Why Not Higher Score:**

As reviewers pointed out, the paper lacks some necessary experiments to strenthen the authors' arguments and therefore needs significant improvements for acceptance.

**Justification For Why Not Lower Score:**

N/A

---

### Decision · Program_Chairs · 2024-01-16

Reject